# Itch/β-arrestin2-dependent non-proteolytic ubiquitylation of SuFu controls Hedgehog signalling and medulloblastoma tumorigenesis

Paola Infante[1], Roberta Faedda[2], Flavia Bernardi[2], Francesca Bufalieri[2], Ludovica Lospinoso Severini[2], Romina Alfonsi[2], Daniela Mazzà[2], Mariangela Siler[2], Sonia Coni[2], Agnese Po[2], Marialaura Petroni[1], Elisabetta Ferretti[3], Mattia Mori [1], Enrico De Smaele[3], Gianluca Canettieri[2,4], Carlo Capalbo[2], Marella Maroder[5], Isabella Screpanti[2], Marcel Kool[6,7], Stefan M. Pfister[6,7,8], Daniele Guardavaccaro [9], Alberto Gulino[2] & Lucia Di Marcotullio[2,4]

Suppressor of Fused (SuFu), a tumour suppressor mutated in medulloblastoma, is a central player of Hh signalling, a pathway crucial for development and deregulated in cancer. Although the control of Gli transcription factors by SuFu is critical in Hh signalling, our understanding of the mechanism regulating this key event remains limited. Here, we show that the Itch/β-arrestin2 complex binds SuFu and induces its Lys63-linked polyubiquitylation without affecting its stability. This process increases the association of SuFu with Gli3, promoting the conversion of Gli3 into a repressor, which keeps Hh signalling off. Activation of Hh signalling antagonises the Itch-dependent polyubiquitylation of SuFu. Notably, different SuFu mutations occurring in medulloblastoma patients are insensitive to Itch activity, thus leading to deregulated Hh signalling and enhancing medulloblastoma cell growth. Our findings uncover mechanisms controlling the tumour suppressive functions of SuFu and reveal that their alterations are implicated in medulloblastoma tumorigenesis.

[1] Center for Life NanoScience@Sapienza, Istituto Italiano di Tecnologia, 00161 Rome, Italy. [2] Department of Molecular Medicine, University La Sapienza, 00161 Rome, Italy. [3] Department of Experimental Medicine, University La Sapienza, 00161 Rome, Italy. [4] Istituto Pasteur-Fondazione Cenci Bolognetti, University La Sapienza, 00161 Rome, Italy. [5] Department of Medico-Surgical Sciences and Biotechnologies, University La Sapienza, 00161 Rome, Italy. [6] Hopp Children's Cancer Center at the NCT (KiTZ), 69120 Heidelberg, Germany. [7] Division of Pediatric Neurooncology, German Cancer Research Center (DKFZ) and German Cancer Consortium (DKTK), 69120 Heidelberg, Germany. [8] Department of Pediatric Hematology and Oncology, Heidelberg University Hospital, 69120 Heidelberg, Germany. [9] Hubrecht Institute-KNAW and University Medical Center Utrecht, 3584CT Utrecht, The Netherlands. These authors contributed equally: Paola Infante, Roberta Faedda, Flavia Bernardi. Correspondence and requests for materials should be addressed to L.D.M. (email: lucia.dimarcotullio@uniroma1.it)

Suppressor of Fused *(SuFu)* is a tumour suppressor gene and negative regulator of Hedgehog (Hh) signalling, a conserved developmental pathway crucial for tissue patterning, stem cell maintenance, and tumorigenesis[1–3]. The SuFu protein is localised to both the nucleus and the cytoplasm and controls the Hh pathway by binding directly to Gli transcription factors, the final effectors of Hh signalling[4,5]. Three Gli proteins have been identified in mammals: Gli1 functions exclusively as a transcriptional activator, whereas Gli2 and Gli3 exist in both full-length (FL) activator and truncated repressor (R) forms. Recently, SuFu has emerged as essential for the stabilisation of Gli2FL and Gli3FL[6], protecting them from degradation by the E3 ubiquitin ligase SPOP. In this context, SuFu regulates the formation of either the repressor or activator forms of Gli3. In the absence of Hh signalling, SuFu restrains Gli3 in the cytoplasm, promoting its processing into the repressor form (Gli3R). Initiation of signalling induces the dissociation of SuFu from Gli3, preventing the formation of Gli3R. This event allows Gli3 to enter the nucleus, where it is converted into a labile transcriptional activator[7]. However, the mechanism by which the SuFu–Gli interaction is controlled remains poorly understood.

SuFu is required for mouse embryonic development[8,9]. Its genetic inactivation leading to constitutive activation of the Hh pathway in a ligand-independent manner causes early embryonic lethality at E9.5 with neural tube defects. In humans, SuFu mutations are associated with Gorlin's syndrome, a hereditary condition characterised by increased risk of developing various forms of tumours, such as basal cell carcinoma and medulloblastoma (MB)[10–12]. Moreover, SuFu is mutated either in the germline or somatically in patients with Sonic hedgehog medulloblastoma (Shh-MB)[10,12–17], a childhood brain tumour associated with Hh signalling aberrations. Despite the central role of SuFu in controlling Hh pathway and its relevance for Hh-dependent tumorigenesis, little information regarding the mechanisms that control its activity is available.

Post-translational modifications, such as phosphorylation and ubiquitylation, affect SuFu stability. Indeed, Shh signalling promotes ubiquitylation of SuFu leading to its proteasomal degradation[18]. This process is opposed by SuFu phosphorylation by glycogen synthase kinase-3β (GSK3β) and cyclic adenosine monophosphate (cAMP)-dependent protein kinase A (PKA) that induce SuFu stabilisation[19]. By means of an enzymatic cascade involving an activating enzyme (E1), a conjugating enzyme (E2), and a ligase (E3) that determines substrate selectivity, ubiquitin is transferred to substrate proteins, generally inducing their degradation by the 26S proteasome[20,21]. Ubiquitin-dependent events have emerged as crucial mechanisms by which stability, activity, or localisation of Gli proteins are controlled[3,22,23]. Gli ubiquitylation is mediated by E3 ligases belonging to the RING-Cullin family, such as Cullin1-Slimb/βTrCP and Cullin3-HIB/Roadkill/ SPOP[24–26], and the HECT family, such as Itch[27,28], as well as by PCAF (P300/CBP-associated factor), a histone acetyltransferase protein with E3 ubiquitin ligase activity[29,30]. Ubiquitylation promoted by these E3 ligases lead to either proteasome-dependent proteolytic cleavage of the Gli2 and Gli3 factors[25,26] or degradation of Gli1[24,27,28]. Although ubiquitylation is a relevant mechanism to control protein degradation[21], it is also required for a variety of non-proteolytic functions.

Here, we identify a new mechanism of regulation of SuFu. We show that the HECT E3 ubiquitin ligase Itch, in complex with the adaptor protein β-arrestin2, binds SuFu and promotes its K63-linked ubiquitylation. This event does not affect SuFu stability. Rather, Itch-mediated ubiquitylation of SuFu facilitates the formation of the SuFu/Gli3 complex, increasing the stability of Gli3FL and, consequently, the amount of Gli3R, thus keeping the Hh pathway off. Moreover, we demonstrate that the Itch-

dependent ubiquitylation of SuFu has a key protective role in MB oncogenesis.

## Results

**The E3 ubiquitin ligase Itch promotes SuFu ubiquitylation**. To identify the molecular mechanisms controlling SuFu activity, we set out to investigate the role of ubiquitylation in the regulation of SuFu. First, we tested whether SuFu can be targeted for ubiquitylation by E3 ubiquitin ligases known to modulate Hh signalling, namely Itch, Nedd4 (HECT E3s), SCF$^{\beta TrCP}$, CRL3$^{SPOP}$, CRL3$^{REN}$, and CRL3$^{KCTD21}$ (Cullin-RING E3s). We found that only Itch was able to ubiquitylate SuFu (Fig. 1a and Supplementary Fig. 1a, b) in cultured cells. Accordingly, increasing amounts of Itch induced a progressive increase in the ubiquitylation of endogenous SuFu (Fig. 1b), while no effect was observed with other HECT E3 ligases (Supplementary Fig. 1a, b). To determine whether SuFu interacts with Itch, we carried out co-immunoprecipitation experiments and demonstrated that Itch interacts with both exogenous (Fig. 1c) and endogenous SuFu (Fig. 1d) in cultured cells. Direct interaction between GST-Itch and in vitro translated [$^{35}$S]-labelled SuFu, as well as between recombinant Itch and GST-SuFu, was also demonstrated by in vitro pull-down assays (Fig. 1e, f). The modular structural organisation of Itch consists of an N-terminal Ca$^{2+}$-dependent phospholipid-binding C2 domain, four WW domains implicated in multiple protein–protein interactions, and a C-terminal catalytic HECT domain (Fig. 1g). To identify the specific domains of Itch involved in the interaction with SuFu, we performed glutathione *S*-transferase (GST) pull-down assays using in vitro transcribed/translated SuFu and different GST-Itch proteins containing only the HECT catalytic domain, the four WW domains, or single WW domains (WW1, WW2, WW3, WW4). We found that the WW1 and WW2 domains of Itch, but not WW3, WW4, or the HECT domain, directly bind to SuFu (Fig. 1h–j).

**Itch ubiquitylates SuFu by K63 linkage**. We next investigated the ability of Itch to ubiquitylate SuFu both in vivo and in vitro. In cultured cells, ectopic expression of Itch, but not of the catalytically inactive ItchC830A mutant, induced the ubiquitylation of endogenous SuFu (Fig. 2a). Moreover, Itch$^{-/-}$ mouse embryonic fibroblasts (MEFs) displayed decreased ubiquitylation of endogenous SuFu when compared to wild-type MEFs (Fig. 2b). SuFu ubiquitylation was rescued by wild-type Itch, but not by its catalytically inactive mutant (ItchC830A) (Fig. 2b).

The Itch-dependent ubiquitylation of SuFu was confirmed in vitro. We used purified recombinant Itch in a reconstituted in vitro ubiquitylation system containing ubiquitin, E1, E2 (UbcH7), adenosine triphosphate (ATP), and in vitro synthesised radiolabelled [$^{35}$S] SuFu as substrate. High levels of SuFu ubiquitylation were observed in the presence of recombinant Itch (Fig. 2c), whereas a SuFu mutant in which all lysine residues were mutated to arginine (SuFu K-less) was not ubiquitylated (Fig. 2d).

To identify the specific lysine residues of SuFu that are ubiquitylated by Itch, we assessed the Itch-dependent ubiquitylation of SuFu mutated in lysine 257, 321, or 457, previously described as direct (K257) or potential (K321 and K457) SuFu ubiquitylation sites[18,31] (Fig. 2e). We observed a significant reduction in the ubiquitylation of SuFuK321R and SuFuK457R mutants, as well as the SuFuK321/457R mutant, when compared to ubiquitylation of wild-type SuFu (Fig. 2f, g and Supplementary Fig. 2). No decrease was observed in the Itch-mediated ubiquitylation of SuFu when lysine 257, previously identified as a ubiquitin acceptor site that induces SuFu proteasomal

degradation[18], was replaced by arginine (K257R) (Supplementary Fig. 2). Overall, these results indicate that Itch ubiquitylates SuFu on lysines 321 and 457.

To determine if Itch-induced ubiquitylation of SuFu primes it for proteasomal degradation, we examined the effect of the proteasome inhibitor MG132 on this event. As shown in Fig. 2h, proteasome inhibition did not cause accumulation of the Itch-dependent ubiquitylation of SuFu. Accordingly, ectopic expression of increasing amounts of Itch or its depletion by RNA interference did not result in any change in the exogenous or endogenous SuFu protein levels, indicating that the Itch-dependent ubiquitylation of SuFu does not lead to SuFu degradation (Fig. 2i–k). Similarly, no changes in the half-life of SuFu were observed in Itch$^{-/-}$ MEFs compared to wild-type MEFs (Fig. 2l).

Next, we analysed the molecular mechanism by which Itch ligates ubiquitin to SuFu. An in vitro ubiquitylation assay performed in the presence of in vitro transcribed/translated SuFu, GST-Itch, and ubiquitin (wild type or mutant) showed that Itch polyubiquitylates SuFu through lysine 63- but not lysine 48-

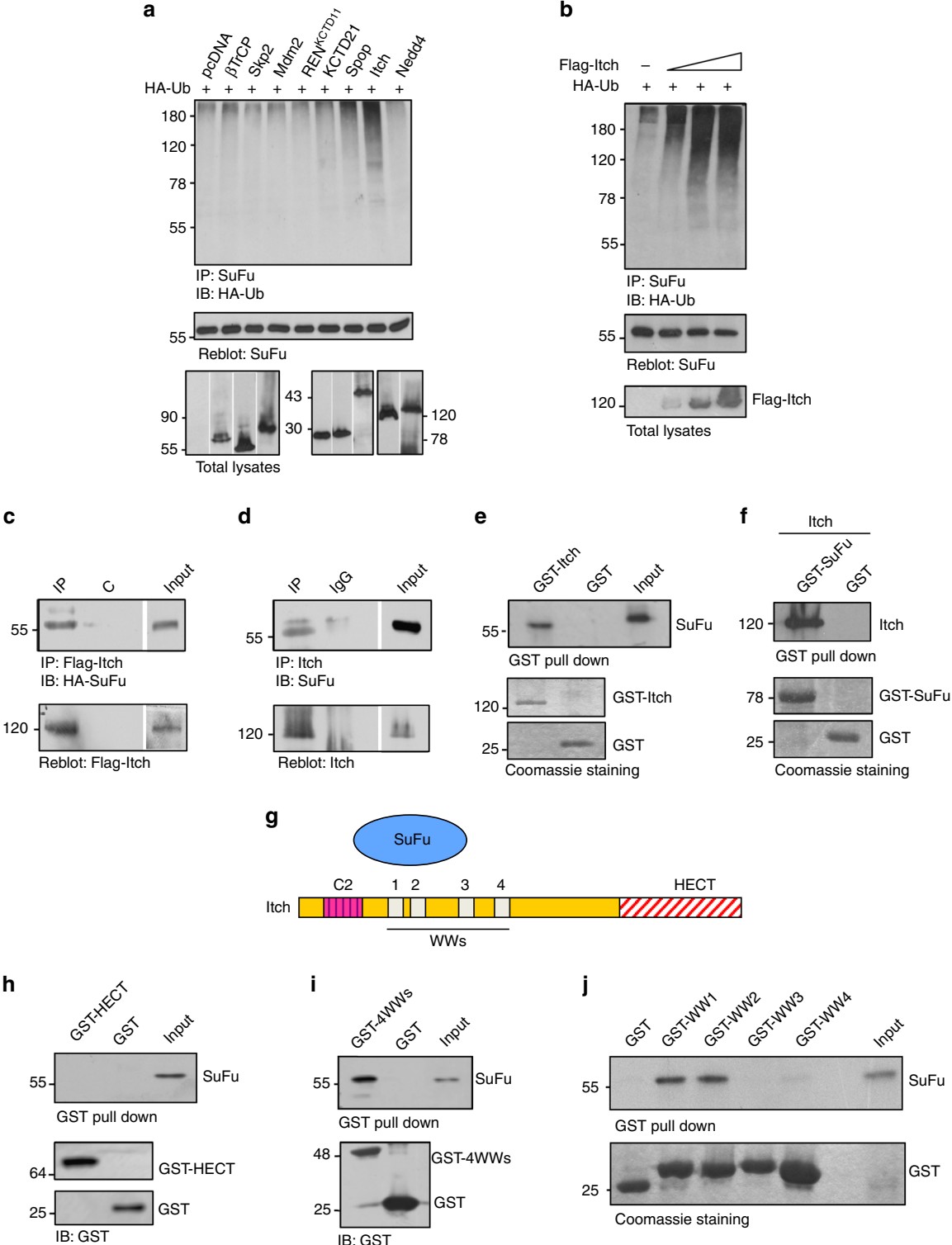

mediated linkages. Indeed, while a ubiquitin mutant in which K48 is replaced by arginine (K48R) was efficiently ligated to SuFu by Itch, a ubiquitin mutant in which K63 is replaced by arginine (K63R) was not linked to SuFu and showed a reduced formation of high polyubiquitin chains and a pattern similar to the one observed in the presence of the K-less ubiquitin mutant, in which all lysine residues are mutated. Accordingly, the ubiquitin mutant containing only lysine 63 (K63O), but not the ubiquitin mutant containing only lysine 48 (K48O), was efficiently ligated to SuFu by Itch, similarly to wild-type ubiquitin (Fig. 2m). Similar results were obtained in in vivo ubiquitylation assays in which the ectopic expression of K63R ubiquitin, but not of the K48R ubiquitin mutant, strongly reduced the Itch-dependent ubiquity-lation of SuFu (Supplementary Fig. 3). Taken together, these results indicate that Itch catalyzes the assembly of K63-linked polyubiquitin chains on SuFu and does not target SuFu for degradation.

**Itch-mediated SuFu ubiquitylation affects SuFu–Gli3 interaction.** K63-linked polyubiquitin chains are thought to serve a scaffolding function for signalling proteins and positively regulate protein complex formation[32]. We hypothesised that the Itch-mediated K63-linked polyubiquitylation could regulate the interaction of SuFu with Gli factors, the final effectors of the Hh pathway. SuFu is known to bind and protect Gli3 from SPOP-mediated degradation, favouring the generation of a cleaved form (Gli3R) that has nuclear repressor activity on Hh-dependent gene expression[7,33].

To determine whether the Itch-dependent ubiquitylation of SuFu increases the association between SuFu and Gli3, we examined the ability of the SuFuK321/457R mutant to interact with Gli3. To this regard we carried out a NanoLucR Binary Technology (NanoBiT) assay, a two-subunit system based on NanoLucR luciferase used for intracellular detection of protein–protein interactions[34,35]. We fused Gli3 and SuFu WT or the SuFuK321/457R mutant protein to Small BiT (SmBiT) or Large BiT (LgBiT) subunits, respectively. The interaction of fusion partners leads to structural complementation of LgBiT with SmBiT, generating a luminescent signal that is a read-out of binding strength. As shown in Fig. 3a, we observed a significant reduction of luminescence when we overexpressed Gli3-SmBiT and SuFuK321/457R-LgBiT mutant compared to wild-type SuFu-LgBiT, demonstrating a decreased association of Gli3 with SuFuK321/457R mutant. Of note, the expression of Itch was able to increase the interaction of Gli3-SmBiT with wild-type SuFu-LgBiT, but not with the SuFuK321/457R-LgBiT mutant.

The effect of Itch-dependent ubiquitylation of SuFu on the formation of the SuFu/Gli3 complex was then tested by immunoprecipitation. Endogenous Gli3 was immunoprecipitated from cells expressing wild-type SuFu or the SuFuK321/457R mutant. The SuFuK321/457R mutant displayed a decreased ability to bind Gli3 when compared to wild-type SuFu, as measured by the absolute levels of SuFu coimmunoprecipitated with Gli3 and by the SuFu/Gli3 ratio (Fig. 3b, c). In agreement with this result, the formation of the SuFu/Gli3 complex was affected by Itch modulation. While Itch depletion by RNA interference caused a reduction of the SuFu/Gli3 interaction (Fig. 3d, e), expression of Itch in Itch$^{-/-}$ MEFs led to increased formation of the SuFu/Gli3 complex (Fig. 3f, g).

To monitor the possible structural consequence of the K321/457R mutation in SuFu, as well as its impact on Gli3 binding affinity, we performed computational studies based on molecular dynamics (MD) simulations and free energy of binding calculations. To this aim, we used the available crystallographic structure of SuFu in complex with a Gli3 peptide (PDB ID: 4BLD) that well represents a static snapshot of the SuFu/Gli3 interaction at high resolution (2.8 Å). The results from this study indicate that the interaction between Gli3 and SuFu is not expected to be impaired by the K321/457R mutation of SuFu at either the structural or thermodynamic level (Supplementary Fig. 4).

We also demonstrated by immunoprecipitation followed by re-immunoprecipitation experiments that SuFu co-purifying with Gli3 is ubiquitylated (Fig. 3h) and that the strong ubiquitylation of SuFu observed in the presence of both Gli3 and Itch was associated with an increased interaction of SuFu with Gli3 (Fig. 3i).

Next, we addressed the role of the Itch-dependent ubiquitylation of SuFu in the regulation of Gli3. As shown in Fig. 3j, in SuFu$^{-/-}$ MEFs the steady-state levels of Gli3FL and Gli3R were higher after the expression of wild-type SuFu than after the expression of the SuFuK321/457R mutant or in control cells. Moreover, in SuFu$^{-/-}$ MEFs the half-life of Gli3FL was significantly shorter after expression of the SuFuK321/457R mutant than after expression of wild-type SuFu (Fig. 3k). In agreement with the finding that SuFu potentiates the formation of Gli3R by controlling the rate of Gli3R production and not the rate of its degradation[7], we found that the half-life of Gli3R remained unchanged. Accordingly, the expression of the SuFuK321/457R mutant resulted in a reduction of Gli3FL and of the Gli3R nuclear fraction when compared to Gli3FL and of the Gli3R nuclear fraction present after the expression of wild-type SuFu (Fig. 3l). The stability of Gli3 was also investigated in response to modulation of Itch. As shown in Fig. 3m, the steady-state levels of Gli3FL and Gli3R were lower after the knockdown of Itch when compared to the ones in control cells. In agreement with this result, the expression of Itch, but not of the ItchC830A mutant, in Itch$^{-/-}$ MEFs led to an increase of Gli3FL stability (Fig. 3n). These data strongly suggest the role of Itch-mediated ubiquitylation of SuFu in

**Fig. 1** Itch ubiquitylates and binds SuFu. **a**, **b** HEK293T cells were transfected with plasmids expressing HA-ubiquitin (HA-Ub) in the presence of different E3 ubiquitin ligases (**a**) or increasing amount of Flag-Itch (**b**). Cell lysates were immunoprecipitated with an anti-SuFu antibody, and ubiquitylated forms were revealed with an anti-HA antibody. **c** HEK293T cells were co-transfected with Flag-Itch and HA-SuFu as indicated. Interaction between Itch and SuFu was detected by immunoprecipitation (IP) followed by immunoblot (IB) analysis with the indicated antibodies. **d** Interaction between endogenous Itch and SuFu was detected in HEK293T cells by immunoprecipitation followed by immunoblot analysis with the indicated antibodies. **e** GST-Itch was bound to glutathione-sepharose beads and used for in vitro pull-down assay. In vitro translated $^{35}$S-labelled SuFu was incubated with free GST control or GST-Itch. After GST pull-down, the protein complex was detected by fluorography. Coomassie blue staining shows the expression levels of recombinant proteins GST-Itch or GST only. **f** GST-SuFu was bound to glutathione-sepharose beads and used for in vitro pull-down assay. Untagged Itch recombinant protein was incubated with free GST control or GST-SuFu. After GST pull-down, the protein–protein interaction was detected by IB with an anti-Itch antibody. Coomassie blue staining shows the expression levels of recombinant proteins GST-SuFu or GST only. **g** Schematic representation of Itch and its interaction with SuFu. **h**, **i** GST-HECT (**h**) or GST-4WWs (**i**) were bound to glutathione-sepharose beads and used for in vitro pull-down assay with in vitro translated $^{35}$S-labelled SuFu. After GST pull-down, protein complexes were analysed by IB. **j** A GST pull-down assay with GST-WW1, -WW2, -WW3, or -WW4 and in vitro translated $^{35}$S-labelled SuFu was carried out as described in **e**

the formation of SuFu/Gli3 complex leading to an increase of Gli3FL stability, and consequently of the Gli3R amount.

Next, we tested whether SuFu ubiquitylation is involved in the negative regulation of Hh-dependent gene expression. Figure 3o shows that overexpression of Gli3 and wild-type SuFu in SuFu$^{-/-}$ MEFs caused a significant reduction of the messenger RNA

(mRNA) levels of Hh target genes. This effect was rescued in the presence of SuFuK321/457R mutant, indicating that Itch-dependent SuFu ubiquitylation is relevant for the suppressive function of SuFu in Hh signalling.

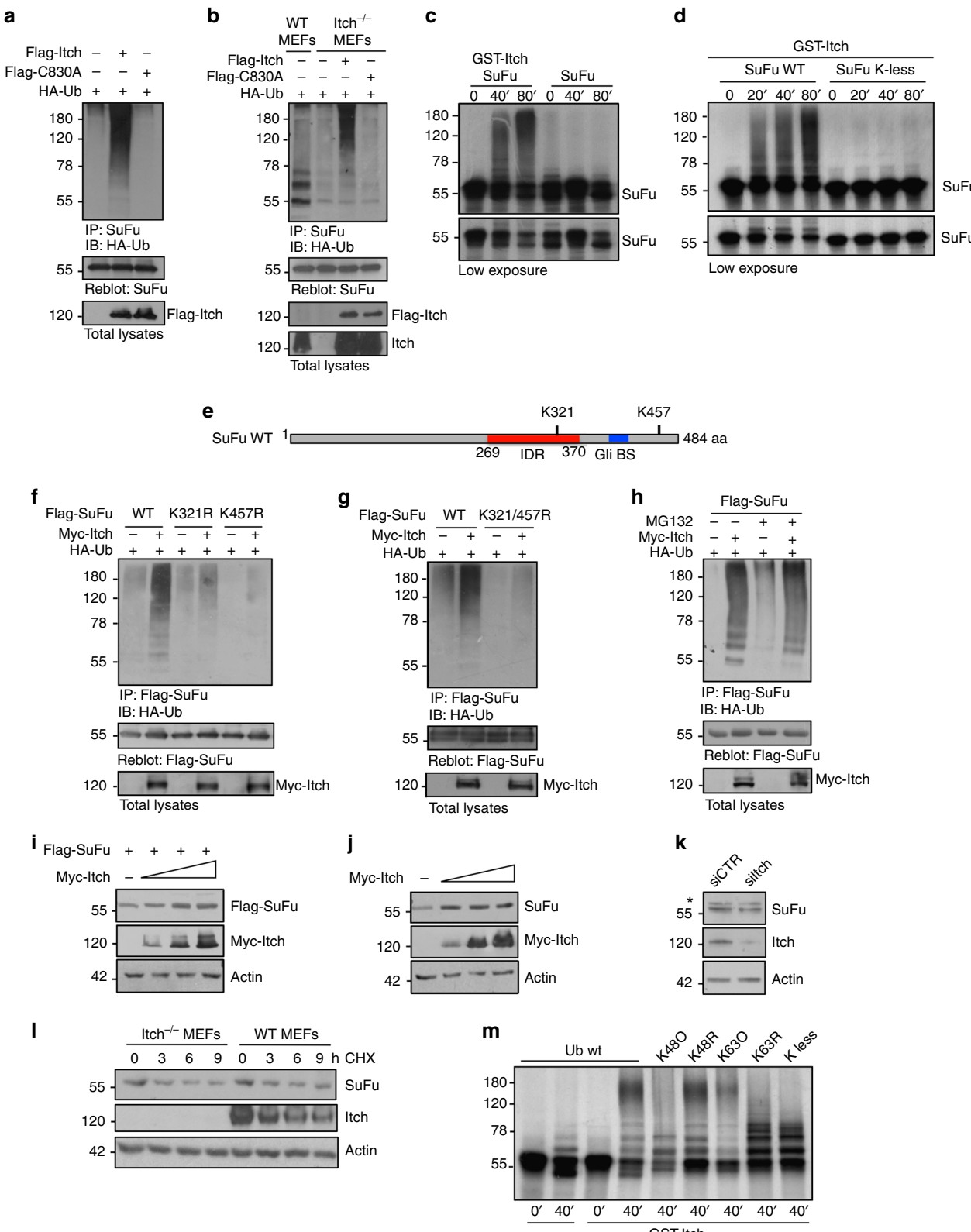

**Hh activation inhibits the Itch-mediated ubiquitylation of SuFu**. To clarify the correlation between SuFu ubiquitylation and the activity of Hh signalling, we first evaluated SuFu ubiquitylation levels during cerebellar development in mice. It is known that Hh signalling regulates cerebellar development by controlling the proliferation of granule cell progenitors (GCPs). During the first week of postnatal development, the cerebellum is formed by an external cortical germinal layer populated by high-proliferating GCPs, which are sustained by high Hh signalling as a result of Purkinje cell-derived Shh stimuli. After the first postnatal week, the physiologic withdrawal of Hh signal causes GCPs growth arrest, their migration in the internal granule layer, and differentiation into mature granules[36,37]. We analysed the ubiquitylation of SuFu and the interaction of SuFu with Gli3 in the postnatal cerebellum tissues of P2- up to P15-old mice and found that SuFu ubiquitylation, as well as the SuFu/Gli3 complex, increased as the pathway is progressively switched off, as indicated by the reduction of the mRNA and protein levels of Gli1, a major read-out of Hh signalling (Fig. 4a–c). These data suggest that SuFu ubiquitylation is functionally related to the activity of Hh signalling. To support this finding, we analysed SuFu ubiquitylation, in the presence or absence of Itch, in NIH3T3 cells treated with Smoothened agonist (SAG), a known small-molecule agonist of the Hh pathway. In agreement with the in vivo results, we observed a reduction of the Itch-dependent ubiquitylation of SuFu in response to Hh signalling activation induced by SAG (Fig. 4d).

Overall, these data support a negative role of the Itch-mediated ubiquitylation of SuFu in the regulation of Hh signalling.

**β-Arrestin2 increases the Itch-mediated ubiquitylation of SuFu**. Itch is an E3 ubiquitin ligase whose activity is regulated by post-translational events such as phosphorylation[38] as well as by proteins that induce its catalytic activity[28,39] and by the interaction with adaptor proteins that mediate the recruitment of specific substrates[40–43]. Of note, among these, the multifunctional adaptor proteins β-arrestins (β-arrestin1 and β-arrestin2) have emerged as important mediators of the Hh pathway[44,45]. Therefore, we tested whether the Itch-dependent ubiquitylation of SuFu was affected by the presence of β-arrestins. We observed that ectopic expression of β-arrestin2, but not β-arrestin1, promoted the ubiquitylation of endogenous SuFu (Fig. 5a). Moreover, expression of β-arrestin2 resulted in increased Itch-mediated ubiquitylation of SuFu as assessed by both in vivo and in vitro ubiquitylation assays (Fig. 5b, c and Supplementary Fig. 5). Accordingly, small interfering RNA (siRNA)-mediated depletion of β-arrestin2 inhibited SuFu ubiquitylation (Fig. 5d).

We also found that β-arrestin2 binds SuFu and that this interaction requires Itch (Fig. 5e).

We then expressed tagged SuFu, Itch, and β-arrestin2 in wild-type MEFs in different combinations. As expected, Flag-tagged Itch co-immunopurified both haemagglutinin (HA)-tagged SuFu and GFP-tagged β-arrestin2. Parallel anti-Flag immunoprecipitates were eluted with Flag peptide and re-immunoprecipitated with an anti-HA antibody. Again, all three proteins were detected by immunoblotting of the second immunoprecipitation indicating that the three proteins, SuFu, Itch, and β-arrestin2, are assembled in a ternary complex (Fig. 5f). Further, this result shows that the presence of β-arrestin2 increases the interaction of SuFu with Itch, as also confirmed by re-expressing β-arrestin2 in βarr2$^{-/-}$ MEFs (Fig. 5g).

Moreover, in agreement with our previous data indicating that SuFu ubiquitylation correlates with the activity of the Hh signalling pathway (Fig. 4), the formation of the SuFu/Itch/β-arrestin2 complex increased as the pathway is progressively switched off, as shown in postnatal mouse cerebellum tissues (Fig. 5h), and was significantly reduced when the pathway was either activated in response to SAG treatment (Fig. 5i, j) or in MB tumour cells in which the Hh pathway is hyperactivated by deletion of Ptch repressor[46] (Fig. 5k).

**Itch-mediated ubiquitylation of SuFu counteracts MB formation**. Several germline and somatic mutations of SuFu have been identified in MB patients[10,12–17]. Recently, whole genome sequencing of a large cohort of Shh-MBs revealed a high frequency of mutations of Hh pathway genes, including new SuFu genetic alterations[47]. Notably, many mutations described so far occur in the C-terminal region of SuFu thus affecting the K321 and K457 residues, suggesting that alterations in Itch-mediated SuFu ubiquitylation might play a key role in MB development.

To address the biological role of the Itch-dependent ubiquitylation of SuFu in the regulation of tumour cell growth, we used human MB Daoy cells belonging to the Shh-MBs subgroup[48–50]. We first compared the proliferation of human MB Daoy cells expressing wild-type SuFu, the SuFuK321/457R mutant, or a control vector. As evaluated by BrdU incorporation and colony-formation assay, wild-type SuFu, but not the SuFuK321/457R mutant, was able to inhibit the proliferation of human MB Daoy cells or decrease the number of colonies (Supplementary Fig. 6a–c). As expected, the SuFuK321/457R mutant retained its inability to block cell growth of Daoy cells following Itch or SuFu modulation (Supplementary Fig. 6e, f). Moreover, we carried out wound healing assays to determine the effect of the Itch-mediated ubiquitylation of SuFu on the migration of MB cells. Daoy cells

**Fig. 2** Itch ubiquitylates SuFu through K63 linkage. **a** HEK293T cells were co-transfected with HA-Ub in the presence or absence of Flag-Itch or Flag-C830A. Cell lysates were immunoprecipitated with an anti-SuFu antibody, followed by immunoblotting with an anti-HA antibody to detect ubiquitylated forms. **b** Itch$^{-/-}$ MEFs were transfected with HA-Ub in the presence or absence of Flag-Itch or Flag-C830A. The assay was carried out as described in **a**. Wild-type (WT) MEFs were used as control to evaluate the basal ubiquitylation of endogenous SuFu. **c, d** In vitro translated $^{35}$S-labelled SuFu WT (**c, d**) or SuFu K-less (**d**) was incubated alone or in combination with GST-Itch for the indicated times. The ubiquitylated forms were detected by fluorography. **e** Schematic representation of SuFu protein showing its lysine residues involved in Itch-dependent ubiquitylation. **f, g** Flag-SuFu WT or Flag-SuFu mutants were co-transfected in HEK293T cells with HA-Ub in the presence or absence of Myc-Itch. The assay was carried out as described in **a**. **h** HEK293T cells were transfected with HA-Ub and Flag-SuFu in the presence or absence of Myc-Itch. Transfected cells were treated with MG132 (50 μM for 4 h) to enrich for ubiquitylated proteins. The assay was carried out as described in **a**. **i** HEK293T cells were transfected with Flag-SuFu in the presence or absence of increasing amount of Myc-Itch. Total protein levels were analysed by immunoblotting. **j** HEK293T cells were transfected in the presence or absence of increasing amount of Myc-Itch. Total protein levels were analysed by immunoblotting. **k** Immunoblotting analysis of SuFu and Itch proteins in HEK293T cells transfected with control (siCTR) or Itch siRNAs (siItch). β-Actin is shown as a control for loading (*non-specific bands). **l** SuFu protein levels in WT MEF or Itch$^{-/-}$ MEF cells treated with cycloheximide (CHX, 100 μg/ml) at different time points. **m** Purified recombinant proteins wild-type Ub or Ub mutants K48 only (K48O), K48R, K63 only (K63O), K63R, or K-less were incubated with GST-Itch and in vitro translated $^{35}$S-labelled SuFu for the indicated times. Ubiquitylated SuFu was detected by fluorography

expressing the SuFuK321/457R mutant displayed an increased motility when compared to cells expressing wild-type SuFu (Supplementary Fig. 6d).

The effect of the SuFuK321/457R mutant on MB cell growth was also tested in vivo. To this end, we xenografted human MB Daoy cells previously infected with control lentiviruses or

lentiviruses carrying wild-type SuFu or the SuFuK321/457R mutant into NOD/SCID mice. Non-invasiveness T2-weighted magnetic resonance imaging (MRI) at 41 days after transplantation showed, as expected, the ability of wild-type SuFu to decrease the tumour volume. Strikingly, this effect was not observed with the SuFuK321/457R mutant (Supplementary Fig. 7a, b).

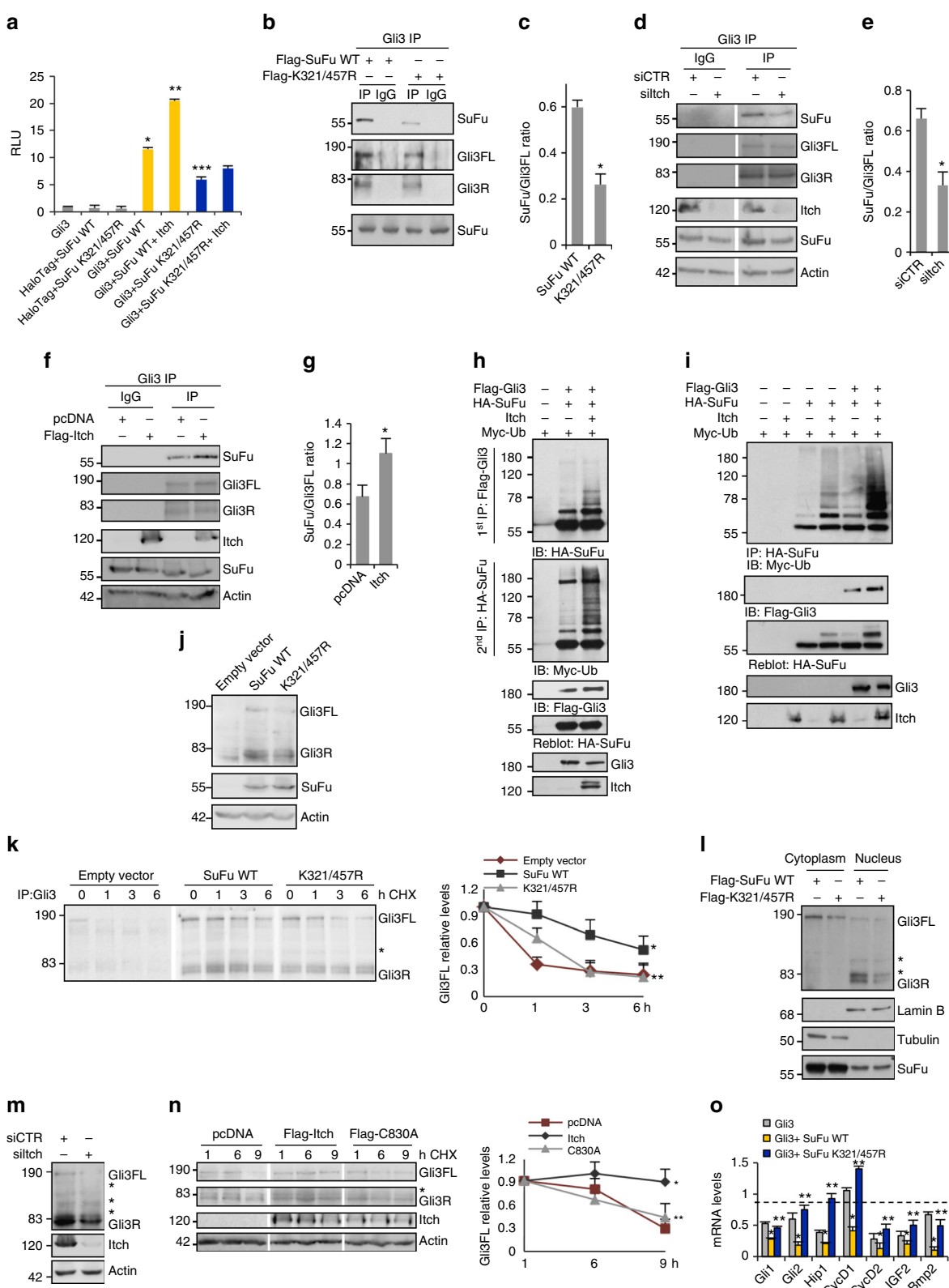

Moreover, a significant decrease of [18]F-fludeoxyglucose (FDG) uptake in mice engrafted with cells expressing wild-type SuFu, but not in mice engrafted with cells expressing SuFuK321/457 R mutant, was observed by PET/SPECT/CT imaging (Fig. 6a) and PET quantification (Fig. 6b). Consistently, compared to mice engrafted with empty lentiviruses, mice engrafted with wild-type SuFu showed a reduced tumour growth rate, a reduced tumour volume (at the end point of experiment) (Fig. 6c–e), a decreased labelling of Ki67 and Gli1 (the final downstream target of Hh signalling) (Fig. 6f, g), and an increased expression of the Gli3 repressor form (Fig. 6h). Notably, all these effects were not observed in mice engrafted with the SuFuK321/457R mutant.

The effect of the SuFuK321/457R mutant on MB cell growth in vivo was confirmed in an MB orthotopic xenograft animal model. Human Daoy cells, previously infected with control lentiviral particles or lentiviral particles expressing wild-type SuFu or the SuFuK321/457R mutant, were implanted into the cerebellum of immunocompromised mice. Assessment of the tumour volume (calculated along serial histologic brain sections) demonstrated the failure of the SuFuK321/457R mutant to decrease the tumour mass as instead observed in wild-type SuFu mice (Fig. 7a, b). Such an effect was likely caused by the inability of the SuFuK321/457R mutant to suppress the tumour cell proliferation as suggested by increased expression of Ki67 and Gli1 observed in mice engrafted with the SuFuK321/457R mutant compared to mice engrafted with wild-type SuFu (Fig. 7c, d).

The relevance of Itch-dependent ubiquitylation of SuFu in the control of tumour growth was also validated in primary MB cells derived from Ptch[+/−] mice[46]. In this mouse model, MB formation results from the deletion of the Ptch gene that leads to constitutive activation of the Hh pathway[46]. Primary MB cells freshly isolated from Ptch[+/−] mice were transduced with control lentiviruses or lentiviruses expressing wild-type SuFu or the SuFuK321/457R mutant. The proliferation of MB cells was impaired by expression of wild-type SuFu, but not by expression of the SuFuK321/457R mutant (Supplementary Fig. 8). Of note, this result was also confirmed in vivo in primary mouse Ptch[+/−] allografts. As expected, compared to mice engrafted with control lentiviruses, mice engrafted with wild-type SuFu showed a reduced tumour growth rate, a reduced tumour volume (at the end point of the experiment), and a decreased Ki67 and Gli1 labelling. Remarkably, these effects were not observed in mice engrafted with the SuFuK321/457R mutant (Fig. 8a–e).

Collectively, these findings demonstrate that the Itch-mediated ubiquitylation of SuFu plays a crucial role for the negative regulation of Hh signalling and explain how alterations of this process, caused by SuFu mutations, contribute to MB oncogenesis.

## Discussion

In the last years, SuFu-Gli3 complex has emerged as a major control node in Hh signalling. However, how the integrity of SuFu–Gli3 complex is maintained and how SuFu is regulated by Hh signalling is still poorly understood.

In the present study, we demonstrate that the HECT E3 ligase Itch in complex with β-arrestin2 ubiquitylates SuFu through K63-mediated linkages. Itch/β-arrestin2-dependent K63-linked poly-ubiquitylation of SuFu on lysines 321 and 457 does not trigger degradation of SuFu; instead, it increases the association of the SuFu-Gli3 complex driving the synthesis of Gli3R, which in turn inhibits signal transduction.

Our findings add further complexity to the regulation of the Hh pathway by ubiquitylation. Indeed, following phosphorylation by PKA, GSK3β, and CK1, which generates binding sites for the SCF[βTrCP] ubiquitin ligase, Gli1 is completely degraded, whereas Gli3 and, to a lesser extent, Gli2, undergo partial proteasomal degradation, leading to the formation of repressor forms that translocate into the nucleus and inhibit the transcription of Hh target genes[51,52]. Downregulation of βTrCP-dependent degradation of Gli proteins is part of an Hh-induced activation signal by which Hh maintains a low degradation mode to enable Gli function[24]. Conversely, HIB/SPOP, the substrate-receptor subunit of the CRL3[HIB/SPOP] ubiquitin ligase, is upregulated by Hh, and promotes Gli2 and Gli3 degradation thus representing an Hh-induced negative feedback loop that modulates signalling activity[6,33,53]. Further, we have recently reported that PCAF induces the proteasome-dependent degradation of Gli1 in response to genotoxic stress[29], and described the ubiquitin-dependent proteolysis of Gli1 mediated by Itch and Numb[27,28]. Itch was also found to regulate the basal turnover of Ptch1[54], thus linking this HECT E3 ligase to different aspects of Hh signalling.

Interestingly, we show that the Itch-induced non-proteolytic ubiquitylation of SuFu is regulated by the adaptor β-arrestin2, a member of the arrestin family of proteins involved in numerous key physiological processes and in cancer progression[55–57]. β-Arrestins have been described to have signalling functions, serve as scaffolds by regulating the internalisation of various types of

**Fig. 3** Itch-dependent K63-linked ubiquitylation of SuFu leads to Gli3R formation. **a** Gli3/SuFu proteins interaction by NanoBiT technology. Itch[−/−] MEFs were transfected with indicated plasmids. *$P < 0.05$, Gli3+SuFu WT versus Gli3; **$P < 0.05$, Gli3+SuFu WT+Itch versus Gli3+SuFu WT; ***$P < 0.05$, Gli3 +SuFuK321/457R versus Gli3+SuFu WT. **b, c** Association between endogenous Gli3 and Flag-SuFu WT or Flag-SuFuK321/457R assayed by determining the amount of SuFu that co-precipitated with anti-Gli3 antibody or control goat antisera (IgG) from MEFs lysates (**b**). The ratio of the SuFu signal to the Gli3FL signal from **b** was plotted (**c**). *$P < 0.05$. **d–g** Gli3/SuFu, interaction, assessed as in (**b**), in WT MEFs transfected with siItch or siCTR (**d**) or in Itch[−/−] MEFs transfected with the indicated plasmids (**f**). The ratio of the SuFu signal to the Gli3FL signal from (**d**) and (**f**) was plotted (respectively (**e**) and (**g**)). *$P < 0.05$. **h** WT MEFs were co-transfected with indicated plasmids in the presence or absence of Itch. Cell lysates were immunoprecipitated with anti-Flag agarose beads (1st IP). After two elutions with Flag peptide, cell lysates were re-immunoprecipitated with anti-HA agarose beads (2nd IP), followed by immunoblotting as indicated. **i** WT MEFs were co-transfected with the indicated plasmids. Cell lysates were immunoprecipitated with anti-HA antibody followed by immunoblotting as indicated. **j** Gli3FL and Gli3R protein levels in SuFu[−/−] MEFs before and after expression of SuFu WT or SuFuK321/457R. **k** Gli3 half-life in SuFu[−/−] MEFs after cycloheximide treatment. Gli3FL and Gli3R protein levels were analysed by immunoprecipitation from whole-cell lysates. The graph shows densitometric analysis. *$P < 0.05$, SuFu WT versus empty vector; **$P < 0.05$, SuFuK321/457R versus SuFu WT. **l** Subcellular fractions generated from WT MEFs transfected with Flag-SuFu WT or Flag-K321/457R. Lamin B and Tubulin were used as nuclear and cytoplasmic markers, respectively. **m** Gli3FL and Gli3R protein levels in WT MEFs transfected with siCTR or siItch. **n** Gli3 half-life in Itch[−/−] MEFs transfected as indicated and then treated with cycloheximide for the indicated times. The graph shows densitometric analysis. *$P < 0.05$, Itch versus pcDNA; **$P < 0.05$, C830A versus Itch. **o** The graphs show the mRNA levels of the indicated Hh target genes in SuFu[−/−] MEFs transfected with Gli3 alone or in combination with Flag-SuFu WT or Flag-K321/457R. *$P < 0.05$, Gli3+SuFu WT versus Gli3; **$P < 0.05$, Gli3+SuFuK321/457R versus Gli3+SuFu WT. *Non-specific band. Each experiment was performed three times independently. Error bars indicate SD. $P$-values were determined using Student's $t$-test

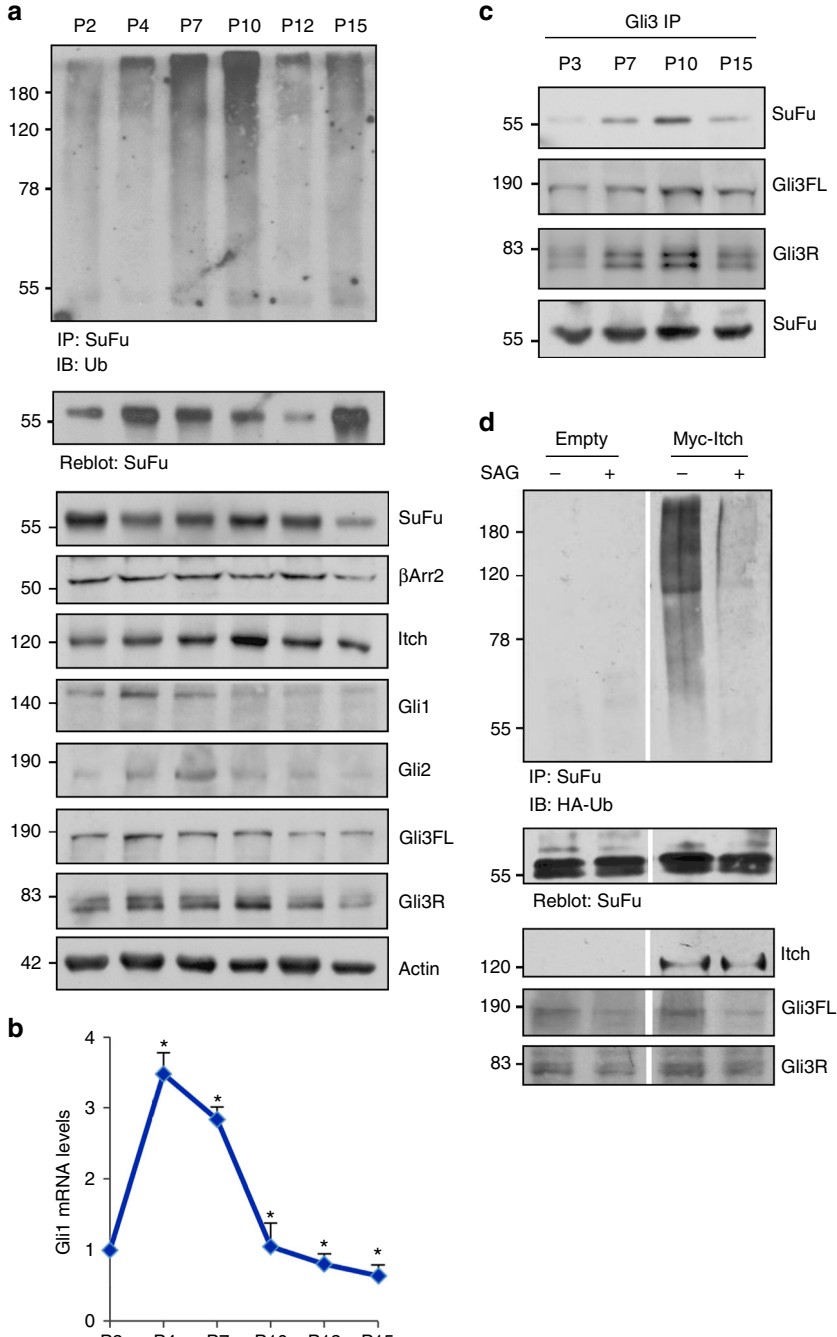

**Fig. 4** Itch-dependent ubiquitylation of SuFu is reverted by Hh pathway activation. **a** Cerebellum lysates from CD1 mice killed at 2d, 4d, 7d, 10d, 12d, and 15d postpartum (P2, P4, P7, P10, P12, P15) were immunoprecipitated with an anti-SuFu antibody and immunoblotted with an anti-Ub antibody. Gli3 proteins were only detected after enriching its levels by immunoprecipitation with an anti-Gli3 antibody. **b** The graph shows the mRNA levels of *Gli1* gene, as a control of pathway activation, in the cerebella described in **a**. Error bars indicate SD from three independent experiments. *$P < 0.05$ (Student's *t*-test). **c** Association between endogenous Gli3 and SuFu from cell lysates of CD1 mice cerebellum (P3, P7, P10, P15) immunoprecipitated with an anti-Gli3 antibody. **d** Itch-dependent SuFu ubiquitylation is inhibited by activation of the Hh pathway. NIH3T3 cells were transfected with HA-Ub in the presence or absence of Myc-Itch and treated with SAG (200 nM for 6 h). Cell lysates were immunoprecipitated with an anti-SuFu antibody and immunoblotted with an anti-HA antibody

receptors, or allow E3 ubiquitin ligase recruitment[58]. Here, we show that expression of β-arrestin2 increases the Itch/SuFu interaction and enhances the Itch-dependent ubiquitylation of SuFu and that, conversely, knockdown of β-arrestin2 inhibits these processes. Biochemical data also demonstrate that β-arrestin2, Itch, and SuFu form a trimeric complex that is promptly dissociated in response to Hh activation. These observations support a new additional role of β-arrestin2 in Hh signalling. β-Arrestin2 has been identified as an important regulator of the Hh pathway that activates signalling by promoting Smo movement to the primary cilium[44,45]. This potential dual role of β-arrestin2 would be similar to that of the type-II kinesin motor protein Kif3a, which, in the absence of Shh, promotes conversion of Gli3 into a truncated repressor form[59], and in the

presence of Shh and in association with β-arrestin2 promotes Gli activator formation by transporting Smo into cilia[45]. We speculate that β-arrestin2 might regulate the switch between pathway OFF, promoting Itch-dependent SuFu ubiquitylation and Gli3R

formation, and pathway ON, triggering the formation of the Smo/β-arrestin2/Kif3a complex.

Mutations in SuFu are a feature of the Shh-MB subgroup. Germline and somatic mutations of SuFu described in patients with medulloblastoma alter SuFu repressor functions. Indeed,

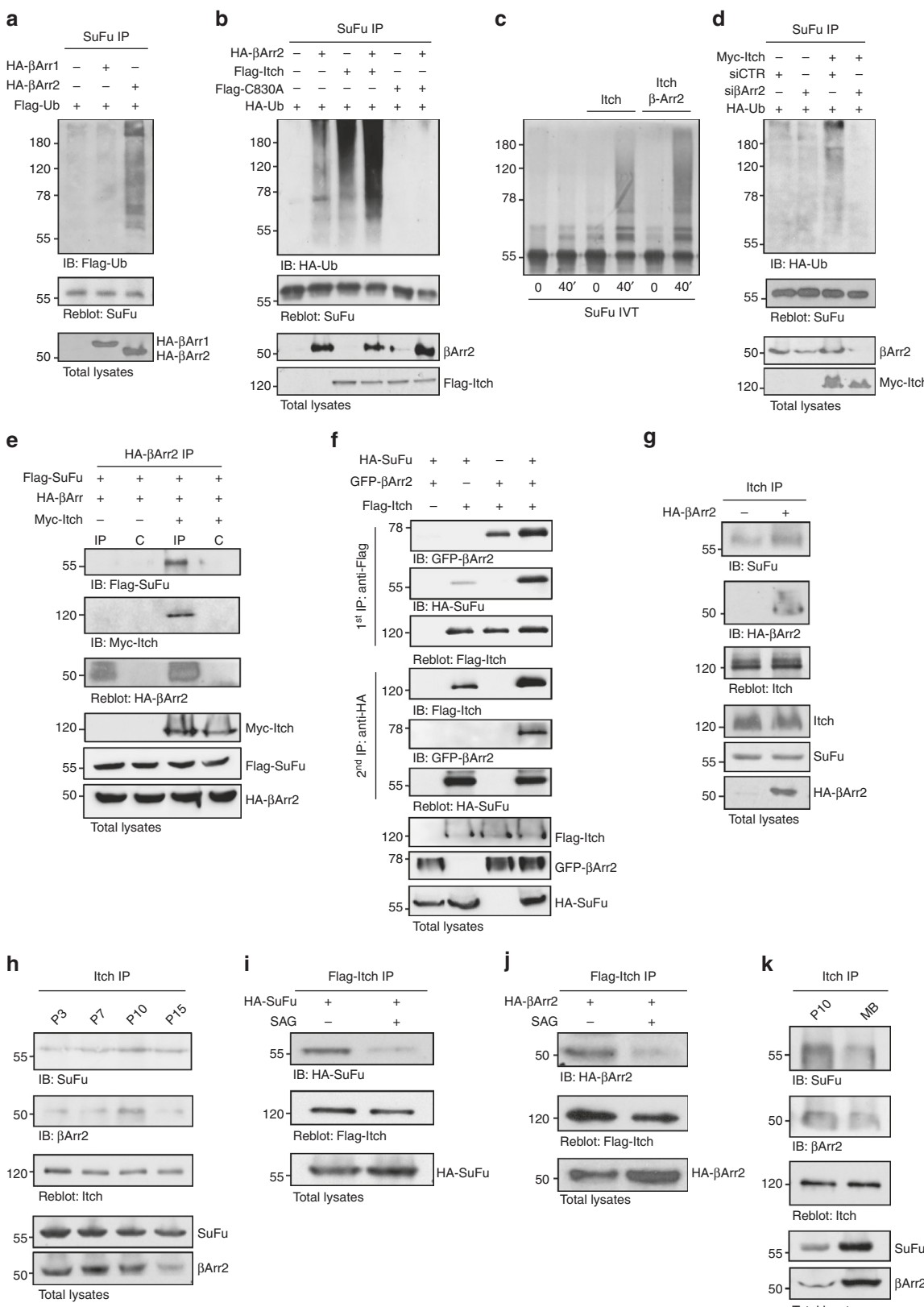

some of these mutations generate truncated proteins that are unable to bind Gli factors[17], thus leading to aberrant activation of Hh signalling. Kool et al.[47] recently reported the genome sequencing of the largest series to date of Shh-related MBs. Interestingly, this group identified two novel non-sense SuFu mutations, Y424X and W430X, both lacking lysine 457 here identified as required for the Itch-dependent ubiquitylation of SuFu (Supplementary Fig. 9a). We have now shown that these mutations encode truncated proteins that conserve Gli consensus, but are unable to be ubiquitylated by Itch (Supplementary Fig. 9b). Accordingly, SuFuY424X or SuFuW430X display a reduced ability to bind Gli3 (Supplementary Fig. 9c), cause a decrease in the abundance of nuclear Gli3R when compared to wild-type SuFu (Supplementary Fig. 9d), and hamper the ability of SuFu to inhibit MB cell growth (Supplementary Fig. 9e). These findings support our model (Fig. 9) that the impairment of SuFu ubiquitylation is implicated in MB oncogenesis.

In conclusion, the Itch-dependent non-proteolytic ubiquitylation of SuFu identified in our study represents a novel mechanism that inhibits the Hh signalling pathway and indicates that alterations of this process, caused by SuFu mutations that make it insensitive to Itch-mediated ubiquitylation, contribute to the pathogenesis of MB.

## Methods

**Plasmids.** pcDNA-Flag-Gli3, pCXN-Flag-REN$^{KCTD11}$, pCXN-Flag-KCTD21, pcDNA-Itch, GST-SuFu, pCHN3HA-SuFu, pCHN3HA-SuFu-W430X and -Y424X constructs were generated in our lab with standard cloning techniques and verified by sequencing. SuFu mutants (SuFuK321/457R, SuFuK321R, SuFuK257R, SuFuK457R) were generated using the QuickChange II site-directed mutagenesis kit (Agilent Technologies, Santa Clara, CA, USA) and verified by sequencing. The following plasmids were kindly provided by other labs: pcDNA-Myc-Itch, pcDNA-Myc-Nedd4, pcDNA-Myc-Ubiquitin (M. Alimandi), pCMV2-Flag-Itch, -Flag-C830A (A. Angers), GST-Itch, GST-HECT, GST-4WWs, GST-WW1, GST-WW2, GST-WW3, GST-WW4, pcDNA-Flag-WWP1 (G. Melino), HA-Ubiquitin constructs (I. Dikic), pRK5-Flag-SuFu (M. Merchant), pcDNA3.1-Flag-SPOP (Wang B), pcDNA-Flag-βTrCP, pcDNA-Flag-Skp2 (M. Pagano). pcDNA3-HA-β-arrestin1, pcDNA3-HA-β-arrestin2, pEGFP-N1-β-arrestin2, and pRK-Myc-Smurf were purchased from Addgene (Cambridge, MA, USA), and pLenti-GIII-CMV-SuFu-GFP-2A-PURO and pLenti-GIII-CMV-GFP-2A-PURO-Blank were purchased from Applied Biological Materials (Richmond, BC, CA).

**Antibodies and reagents.** Rabbit anti-SuFu C81H7 (#2522S, 1:3000), rabbit anti-β-arrestin2 C16D9 (#3857S, 1:1000), and mouse anti-Gli1 L42B10 (#2643S, 1:500) were from Cell Signalling (Beverly, MA, USA); goat anti-SuFu C-15 (sc-10933 2 μg), mouse anti-GST B-14 (sc-138, 1:10,000), goat anti-Actin I-19 (sc-1616, 1:1000), goat anti-Lamin B M-20 (sc-6217, 1:1000), mouse anti-α Tubulin TU-02 (sc-8035, 1:1000), mouse anti-Ub P4D1 (sc-8017, 1:500), rabbit anti-Gli1 H300 (sc-20687, 1:100), mouse anti-HA-probe F-7 horseradish peroxidase (HRP) (sc-7392 HRP, 1:1000), mouse anti-c-Myc 9E10 HRP (sc-40 HRP, 1:500), mouse anti-GFP (B-2) HRP (sc-9996 HRP, 1:1000), and HRP-conjugated secondary antibodies were purchased from Santa Cruz Biotechnology (Santa Cruz, CA, USA); anti-Flag M2 HRP (A8592, 1:1000), anti-Flag M2 agarose (A2220, 1–2 μg), and anti-HA agarose (A2095, 1–2 μg) were from Sigma Aldrich (St Louis, MO, USA); goat anti-Gli3 (AF3690, 1:1000 or 2–4 μg) and goat anti-Gli2 (AF3635, 1:1000) were from R&D Systems (Minneapolis, MN, USA); mouse anti-Itch (611199, 1:2000) antibody was purchased from BD Bioscience (Heidelberg, Germany); rabbit anti-Ki67 SP6 (MA5-14520, 1:100) was from Thermo Fisher Scientific (Waltham, MA, USA); mouse anti-MDM2 OP46 (Ab-1, 1:500) was from Calbiochem (Darmstadt, Germany). Where indicated, cells were treated with SAG (200 nM, Alexis Biochemicals Farmingdale, NY, USA) for 6 h, MG132 (50 μM; Calbiochem, Nottingham, UK) for 4 h, or Cycloheximide (CHX 100 μg/ml, Sigma Aldrich).

**Cell culture and transfection and lentiviral infection.** HEK293T cells, wild-type MEFs, SuFu$^{-/-}$, Itch$^{-/-}$, and βArr2$^{-/-}$ MEFs were cultured in Dulbecco's modified Eagle's medium plus 10% fetal bovine serum (FBS) or 10% bovine serum (BS) for NIH3T3 cells. Daoy cells were maintained in Eagle's minimum essential medium (MEM) plus 10% FBS. All media contained L-glutamine and antibiotics.

Primary MB cells were freshly isolated from Ptch$^{+/-}$ mice as previously described[60]. Cells were cultured in Neurobasal Media-A with B27 supplement minus vitamin A[61]. HEK293T (CRL-3216™), NIH3T3 (CRL-1658™), and Daoy (HTB-186™) cells were obtained from ATCC. Itch$^{-/-}$ MEFs were a gift from Dr C. Brou (Institut Pasteur), SuFu$^{-/-}$ MEFs were a gift from Dr R. Toftgård (Karolinska Institutet), and βArr2$^{-/-}$ MEFs were a gift from Dr R.J. Lefkowitz (Howard Hughes Medical Institute).

Mycoplasma contamination in cell cultures was routinely detected by using PCR detection kit (Applied Biological Materials, Richmond, BC, Canada).

Transient transfections were performed using DreamFect™Gold transfection reagent (Oz Biosciences SAS, Marseille, FR) or Lipofectamine® with Plus™ Reagent (Thermo Fisher Scientific, Waltham, MA, USA) in accordance with the manufacturer's protocols. For lentiviral infection, HEK293 cells were transfected with lentiviral constructs and the packaging plasmids pMD.G and pCMVR8.74 using Calcium/Phosphate precipitation. Culture medium containing the lentivirus was collected 48 and 72 h after transfection. Daoy cells were infected with purified lentiviruses for 48 h in the presence of 4 μg/ml polybrene (Sigma Aldrich, St Louis, MO), primary MB cells were infected with purified lentiviruses for 48 h, and SuFu$^{-/-}$ MEFs were infected with lentiviruses in the presence of 8 μg/ml polybrene for 72 h.

For RNA interference, scrambled (Cat no: D-001810-10-05), Itch (Cat no: L-007196-00-0005), SuFu (Cat no: E-015382-00), or β-arrestin2 (Cat no: J-041022-11 and J-041022-10) short interfering RNA oligos (Dharmacon, Inc., Lafayette, CO, USA) were transfected for 48 h using Lipofectamine 2000® (Thermo Fisher Scientific) or HiPerFect Transfection Reagent (Qiagen, Milan, IT).

**GST pull-down assay.** Recombinant GST-fusion proteins were expressed in *Escherichia coli* BL21, and purified as previously described[62]. GST, GST-SuFu, and GST-Itch recombinant proteins were bound to glutathione beads (GE Healthcare, Pittsburgh, PA, USA) and incubated for 2 h with in vitro translated protein in binding buffer (4-(2-hydroxyethyl)-1-piperazineethanesulfonic acid 20 mM, MgCl$_2$ 2 mM, KCl 100 mM, 20% glycerol, EDTA 0.2 mM, 0.05% NP-40) and analysed by immunoblotting or fluorography. As an alternative, GST or GST-SuFu recombinant proteins bound to glutathione beads were incubated for 2 h with untagged

**Fig. 5** β-arrestin2 increases the Itch-dependent SuFu ubiquitylation. **a** WT MEFs were transfected with Flag-Ub in the presence or absence of HA-β-arrestin1 (HA-βArr1) or HA-β-arrestin2 (HA-βArr2). Cell lysates were immunoprecipitated with an anti-SuFu antibody and ubiquitylated forms were revealed with an anti-Flag antibody. **b** HEK293T cells transfected with the indicated plasmids were immunoprecipitated with an anti-SuFu antibody. Ubiquitylated forms were revealed with an anti-HA antibody. **c** In vitro translated $^{35}$S-labelled SuFu was incubated alone or in combination with untagged recombinant Itch protein in the presence or absence of recombinant β-arrestin2 protein for the indicated times. Levels of ubiquitylated $^{35}$S-labelled SuFu were detected by fluorography. **d** SuFu ubiquitylation in WT MEFs transfected with HA-Ub in the presence or absence of Myc-Itch and with specific siRNA for β-arrestin2 (siβArr2) or non-specific control (siCTR). Immunoprecipitation and immunoblotting were performed as in (**b**). **e** HEK293T cells were co-transfected with the indicated plasmids. Interaction of β-arrestin2 with SuFu and Itch was detected by immunoprecipitation followed by immunoblot analysis with the indicated antibodies. **f** SuFu, Itch, and β-arrestin2 form a trimeric complex. WT MEFs were transfected with different combinations of Flag-Itch, GFP-β-arrestin2, and HA-SuFu constructs. Protein lysates were immunoprecipitated with anti-Flag agarose beads. One-third of immunocomplexes was probed with antibodies to the indicated proteins (1st IP), whereas two-thirds were subjected to two elutions with Flag peptide and re-immunoprecipitated with HA-agarose beads followed by immunoblotting as indicated (2nd IP). **g** β-Arrestin2$^{-/-}$ MEF cells were transfected with HA-β-arrestin2 plasmid. Interaction of Itch with SuFu and β-arrestin2 was detected by immunoprecipitation followed by immunoblot analysis with the indicated antibodies. **h** Association of endogenous Itch with SuFu and β-arrestin2 from cell lysates of CD1 mice cerebellum (P3, P7, P10, P15) immunoprecipitated with an anti-Itch antibody. **i, j** NIH3T3 cells transfected with the indicated plasmids were treated with SAG (200 nM for 6 h) or vehicle only. Interaction between Flag-Itch and HA-SuFu (**i**) or HA-β-arrestin2 (**j**) was detected by anti-Flag immunoprecipitation, followed by immunoblot analysis with the indicated antibodies. **k** Association of endogenous Itch with SuFu and β-arrestin2 from cell lysates of CD1 mice cerebellum (P10) and from MB Ptch$^{+/-}$ tissue both immunoprecipitated with an anti-Itch antibody

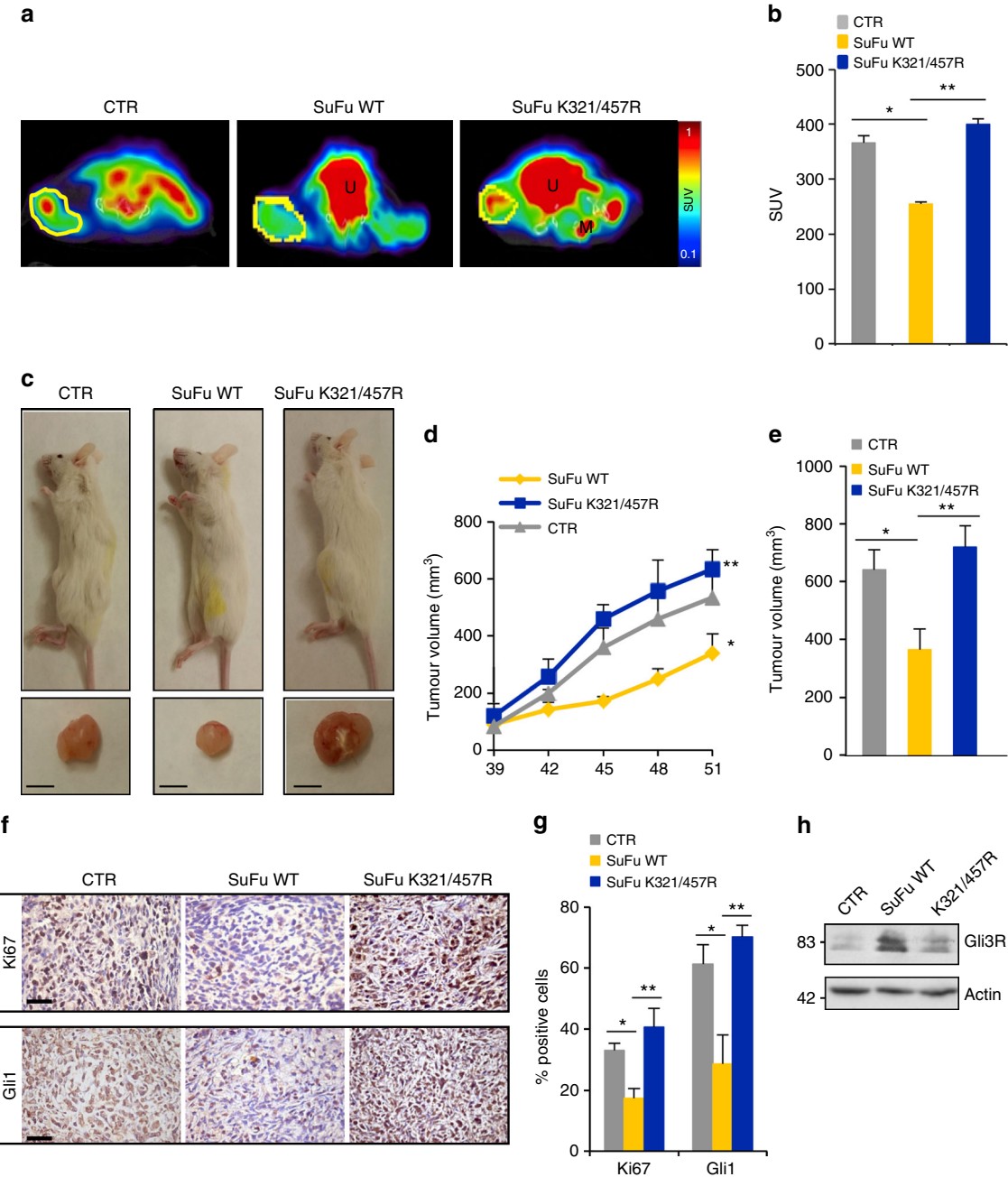

**Fig. 6** Human Daoy MB xenografts. **a** PET/CT images of representative CTR ($n = 6$), SuFu WT ($n = 6$), and SuFu K321/457R ($n = 8$) mice showing tumour FDG (F-18-fluorodeoxyglucose) uptake at 41 days after implantation. ROI (region of interest) drafts the tumour mass. Significant differences were observed between SuFu WT and SuFu K321/457R mutant mice. U urinary bladder, M femoral muscle. **b** Graphic representation of SUV (standard uptake value). For each tumour, the SUV as mean tumour FDG uptake normalised for animal body weight was calculated. Significant difference in tumour FDG uptake was observed between SuFu K321/457R and SuFu WT mice. *$P < 0.05$, SuFu WT versus CTR; **$P < 0.05$, SuFu K321/457R versus SuFu WT. **c** Images of xenografted NOD/SCID mice with bilateral MB xenograft tumours. Tumour volume in SuFu K321/457R mutant mice is visually bigger than SuFu WT (scale bars = 5 mm). **d** Tumour volumes were monitored over time by caliper measurements at the indicated times. *$P < 0.01$, SuFu WT versus CTR; **$P < 0.01$, SuFu K321/457R versus SuFu WT. **e** Tumour volumes were measured post explantation. *$P < 0.01$, SuFu WT versus CTR; **$P < 0.01$, SuFu K321/457R versus SuFu WT. **f** Immunohistochemistry of Ki67 and Gli1 stainings. Scale bars indicate 50 µm. **g** Quantification of Ki67 and Gli1 stainings from immunohistochemistry. *$P < 0.01$, SuFu WT versus CTR and **$P < 0.01$, SuFu K321/457R versus SuFu WT. **h** Western blot analysis shows protein expression levels. Error bars indicate SD. $P$-values were determined using Mann–Whitney $U$-test

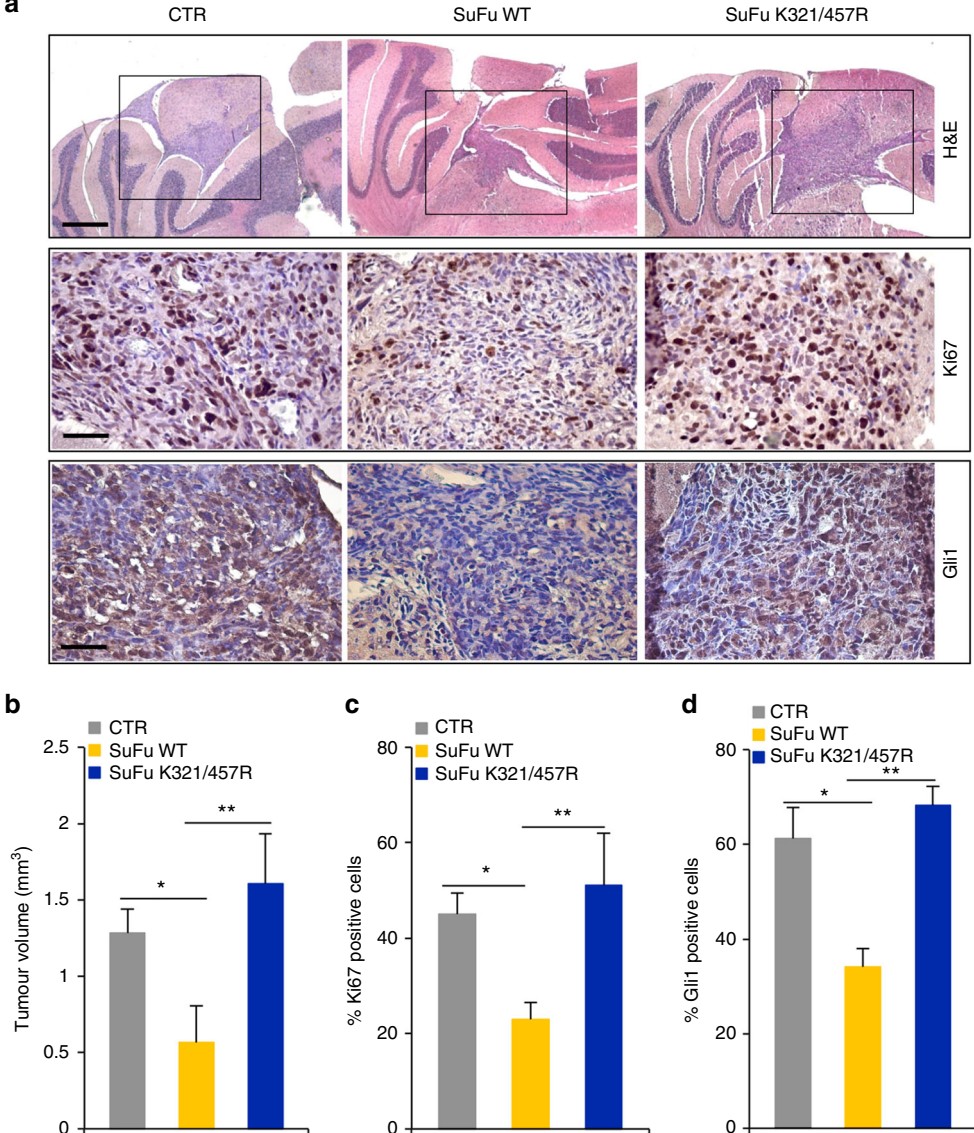

**Fig. 7** Human Daoy orthotopic MB xenografts. **a** Representative images of haematoxylin and eosin (H&E), Ki67, and Gli1 immunohistochemical stainings of a human Daoy MB cell-derived orthotopic tumour in NOD/SCID mice (n = 6 mice for each group, CTR, SuFu WT, and SuFu K321/457R). Scale bars represent 500 μm for H&E staining and 50 μm for Ki67 and Gli1 stainings. **b** Representative tumour average volumes after explantation. **c**, **d** Quantification of Ki67 (**c**) and Gli1 (**d**) stainings from immunohistochemistry shown in (**a**). *P < 0.01, SuFu WT versus CTR; **P < 0.01, SuFu K321/457R versus SuFu WT. Error bars indicate SD. P-values were determined using Mann–Whitney U-test

recombinant protein Itch (Boston Biochem Cambridge, MA, USA) in binding buffer and analysed by immunoblotting.

**Immunoblot analysis and immunoprecipitation.** Transfected cells were lysed in a solution containing 50 mM Tris at pH 7.4, 300 mM NaCl, 2% NP-40, 0.25% deoxycholic acid sodium, 1 mM dithiothreitol (DTT), and protease inhibitors or in RIPA buffer (50 mM Tris at pH 7.6, 150 mM NaCl, 0.5% deoxycholic acid sodium, 5 mM EDTA, 0.1% SDS, 100 mM NaF, 2 mM NaPPi, 1% NP-40). The lysates were centrifuged at 13,000×g for 30 min and the resulting supernatants were subjected to immunoblot analysis or immunoprecipitation. For immunoprecipitation, cell lysates were immunoprecipitated with specific antibodies: anti-Flag and anti-HA agarose or anti-Gli3, anti-SuFu, and anti-Itch antibodies from 2 h to overnight at 4 °C with rotation. Flag- or HA-peptide (0,1 mg/ml, Sigma Aldrich) or IgG (1–2 μg, Santa Cruz Biotechnology) were used as a control. The immunoprecipitations performed by the use of primary antibodies not conjugated to agarose beads were followed by incubation with Protein G- or Protein A-agarose beads (Santa Cruz Biotechnology) for 1 h with rotation[63]. The immunoprecipitates were then washed five times with the lysis buffer described above, resuspended in sample loading buffer, boiled for 5 min, resolved in sodium dodecyl sulphate–polyacrylamide gel electrophoresis (SDS-PAGE), and then subjected to immunoblot analysis.

Uncropped images of the most important blots were shown in the Supplementary Figures.

**Subcellular fractionation.** Subcellular fractionation experiments were performed as previously described[7]. Briefly, freshly collected cells were washed twice with phosphate-buffered saline (PBS) and twice with 10 mM HEPES (pH 7.4) and then were incubated for 10 min in 10 mM HEPES (pH 7.4). Subsequently, the cells were lysed in SEAT Buffer (10 mM triethanolamine/acetic acid at pH 7.4, 250 mM sucrose, 1× EDTA protease inhibitor cocktail) by 15 passages through a 25-G needle. The lysates were centrifuged at 900×g for 5 min and the resulting supernatants were brought to 1× buffer A (50 mM Tris at pH 7.4, 300 mM NaCl, 2% NP-40, 0.25% deoxycholic acid sodium, 1 mM DTT, and protease inhibitors), extracted for 1 h, and clarified by centrifugation at 20,000×g for 1 h to obtain the citoplasmatic fraction. On the other side, the nuclear pellets were washed once in SEAT buffer. The nuclei were extracted for 1 h with 20 mM HEPES at pH 7.9, 1 mM MgCl₂, 0.5 M NaCl, 0.2 mM EDTA, 20% glycerol, 1% Triton X-100, 1 mM DTT, benzonase, and a protease inhibitor cocktail and clarified by centrifugation at 20,000×g for 1 h. An equal percentage of nuclear and cytoplasmic fractions were loaded for lane to ensure that equal amounts of each fraction were loaded on the gel.

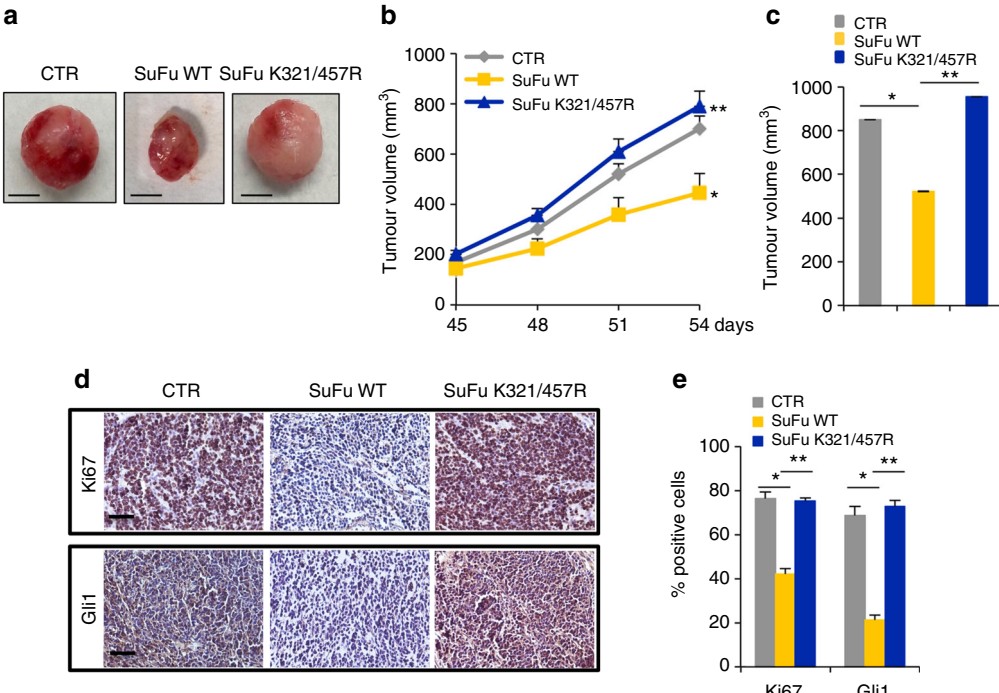

**Fig. 8** Mouse Ptch1$^{+/−}$ MB allografts. **a** Images of representative CTR, SuFu WT, and SuFu K321/457R flank allograft masses ($n = 6$ mice for each group) (scale bars = 5 mm). **b** Tumour volumes were monitored over time by caliper measurements at the indicated times. *$P < 0.05$, SuFu WT versus CTR; **$P < 0.05$, SuFu K321/457R mutant versus SuFu WT. **c** Tumour volumes were measured post explantation. *$P < 0.05$, SuFu WT versus CTR; **$P < 0.05$, SuFu K321/457R mutant versus SuFu WT. **d** Immunohistochemistry of Ki67 and Gli1 stainings. Scale bars indicate 50 μm. **e** Quantification of Ki67 and Gli1 stainings from immunohistochemistry. *$P < 0.05$, SuFu WT versus CTR and **$P < 0.05$, SuFu K321/457R mutant versus SuFu WT, for Ki67 staining quantification. *$P < 0.05$, SuFu WT versus CTR and **$P < 0.01$, SuFu K321/457R versus SuFu WT, for Gli1 staining quantification. Error bars indicate SD. $P$-values were determined using Mann–Whitney $U$-test

**In vivo ubiquitylation assay**. In vivo ubiquitylation experiments were performed as previously described[29]. MEFs or HEK293T cells transfected with various plasmids were lysed with denaturing buffer (1% SDS, 50 mM Tris at pH 7.5, 0.5 mM EDTA, 1 mM DTT) to disrupt protein–protein interactions and then lysates were diluted 10 times with lysis buffer and subjected to immunoprecipitation with antibodies indicated in figures for 2 h to overnight at 4 °C. The immunoprecipitated proteins were then washed five times with the lysis buffer described above, resuspended in sample loading buffer, boiled for 5 min, resolved in SDS-PAGE, and then subjected to immunoblot analysis. Polyubiquitylated forms were detected using anti-HA, anti-Flag, anti-Myc, or anti-Ubiquitin antibodies.

**In vitro ubiquitylation assay**. In vitro ubiquitylation was performed as previously described[64,65]. In vitro translated protein SuFu, produced using TnT® Coupled Wheat Germ Extract System (Promega, Madison, WI, USA), was incubated at 30 °C with GST or Itch-GST or Itch-GST and βarr2-GST 400ng (Abnova, Heidelberg, Germany), 50 mM Tris-HCl at pH 7.5, 5 mM MgCl$_2$, 200 μM okadaic acid, 2 mM ATP, 0.6 mM DTT, 1 mM ubiquitin aldehyde, E1, UbcH7, and either wild-type or mutant ubiquitin (Boston Biochem). Polyubiquitylated products were subjected to SDS-PAGE and detected by fluorography.

**NanoLuc Binary Technology assay**. Gli3, SuFu WT, and SuFuK321/457R were cloned into vectors compatible with the Flexi Vector System (NanoBiT™, Promega) in accordance with the manufacturer's protocols. For our assay we used Gli3 protein fused to the SmBiT subunit and SuFu WT or SuFuK321/457R mutant protein fused to LgBiT subunit. Itch$^{−/−}$ MEFs cells were seeded in 96-well plates and transfected with the plasmids or with the NanoBiT Negative Control Vector, which encodes HaloTag-SmBiT used in combination with SuFu WT-LgBiT or SuFu K321/457R-LgBiT. At 24 h post transfection, the Nano-Glo Live Cell Reagent was added to a 1× concentration and luminescence was measured using the Glo-Max Multi+Detection System (Promega).

**mRNA expression analysis**. Total RNA was isolated with TRIzol (Thermo Fisher Scientific) and reverse-transcribed with SensiFASTcDNA Synthesis Kit (Bioline Reagents Limited, London, UK). Quantitative real-time PCR (Q-PCR) analysis of *Gli1*, *Gli2*, *Hip1*, *CyclinD1*, *CyclinD2*, *Igf2*, and *Bmp2* mRNA expression was performed on each complementary DNA (cDNA) sample using the ViiA™ 7 Real-Time PCR System (Life Technologies, Foster City, CA, USA). A reaction mixture containing cDNA template, SensiFAST™ Probe Lo-ROX mix (Bioline Reagents Limited), and primer probe mixture was amplified using standard Q-PCR thermal cycler parameters. Each amplification reaction was performed in triplicate and the average of the three threshold cycles was used to calculate the amount of transcript in the sample (using SDS version 2.3 software). mRNA quantification was expressed, in arbitrary units, as the ratio of the sample quantity to the calibrator. All values were normalized with two endogenous controls, *β-2 microglobulin* and *HPRT*, which yielded similar results.

**Cell proliferation and wound healing assays**. Cell proliferation was evaluated by BrdU detection (Roche, Welwyn Garden City, UK). Briefly, after the BrdU pulse cells were fixed with 4% paraformaldehyde and permeabilized with 0.2% Triton X-100, and BrdU detection was performed according to the manufacturer's instructions. Nuclei were counterstained with Hoechst reagent. At least 500 nuclei were counted in triplicate, and the number of BrdU-positive nuclei was recorded. For colony-formation assays, $1 \times 10^4$ Daoy cells, infected with lentiviral particles, were plated in 10-cm-diameter dishes, and after 2 weeks of puromycin (Sigma Aldrich) selection, cell colonies were counted in triplicate after staining in 20% methanol and crystal violet. Cells were counted in triplicate. For wound healing assay Daoy cells, were infected with lentiviral particles and then seeded at high density into 12-multiwell plates. The following day, a linear scratch in the confluent cell monolayer was made with a sterile pipette tip. Cells were washed three times with PBS with Ca$^{2+}$ and Mg$^{2+}$ and incubated in regular media. For each well, three pictures were taken along the scratch area at the indicated times, and the wound areas were calculated using ImageJ. Cell migration was defined as the reduction of the wound area in each photographed field during the course of the treatment.

**Animal studies**. For xenograft models, $4 \times 10^6$ Daoy cells infected with lentiviral particles were resuspended in an equal volume of MEM medium and Matrigel (BD Biosciences, Heidelberg, Germany) and injected subcutaneously (s.c.) at the posterior flank of 6-week-old female NOD/SCID mice ($n = 6$ CTR, $n = 6$ SuFu WT, $n = 8$ SuFuK321/457R) (Charles River Laboratories, Lecco, Italy). After 40 days following tumour cells injection, animals were imaged using a dedicated whole body mouse coil in a 1T MRI scanner (Bruker, Icon, Germany). Mice were anaesthetised with 1–2% isofluorane in air and O$_2$. During measurements,

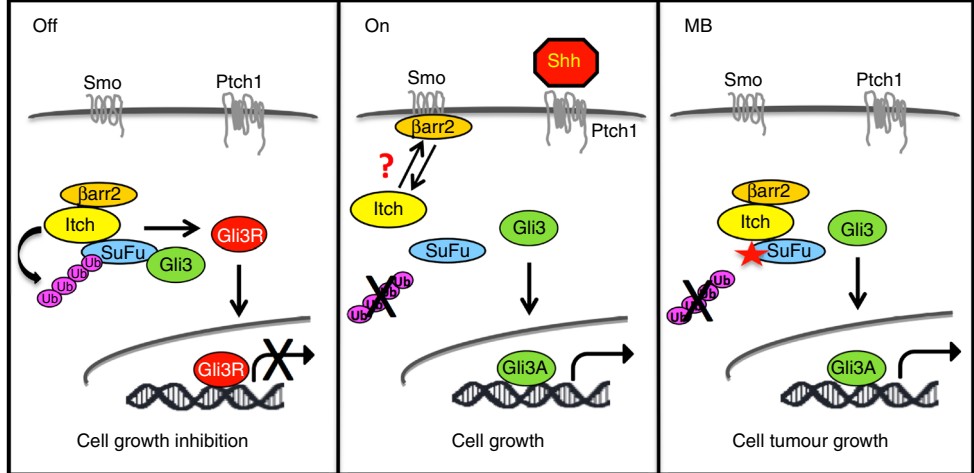

**Fig. 9** Model of the Itch/β-arrestin2-dependent regulation of the SuFu/Gli3 complex function. When Hh pathway is OFF, Itch, coadjuvated by the protein adaptor β-arrestin2, ubiquitylates SuFu. This event does not lead to SuFu degradation, but increases the association between SuFu and Gli3. In this way Gli3 is protected from SPOP-dependent degradation and is cleaved into a repressor form (Gli3R) that inhibits Hh target gene transcription and cell growth. When Hh pathway is switched ON, β-arrestin2 dissociates from the SuFu/Itch complex, thus abrogating Itch-dependent SuFu ubiquitylation. This process induces the dissociation of the Gli3-SuFu complex and impairs Gli3R formation, thereby leading to Hh pathway activation and sustained cell growth. Alterations in this mechanism, caused by SuFu mutations that make it insensitive to Itch-dependent ubiquitylation, are responsible for MB tumorigenesis

respiration was monitored with a sensor connected to an ECG/respiratory unit. T2-weighted MR scans with Fast Spin-Echo RARE sequence (TR/TE = 2500/35 ms, number of average (NA = 2)), were acquired for the tumour volume evaluation. Animals were imaged with a PET/SPECT/CT scanner (Trifoil Imaging, USA) equipped with gas anaesthesia, respiratory monitoring, and heated bed. Mice were injected with 9.43 MBq (SD 1.26; range 7.43–11.61 MBq) of 18F-FDG (AAA, Italy). PET imaging started 45 min after FDG injection and lasted 30 min. Positron emission tomography (PET) images were corrected for decay, randoms, dead time, and computed tomography (CT) attenuation, and reconstructed using an OSEM algorithm with 8 subsets and 30 iterations. PET quantitative data were obtained by an investigator blinded to the group identity by outlining the tumours directly on the anatomical CT images. At the end of PET experiment, animal were killed with an overdose of anaesthetic, the tumours were collected, and the volume ($mm^3$) of each tumour measured with a caliper and calculated by the formula ($length^2 \times$ width) / 2 where length refers to the smaller dimension.

For orthotopic xenografts model, Daoy cells (prepared from fresh culture to ensure optimal viability of cells during tumour inoculation) infected with lentiviruses particles were stereotaxically implanted ($2 \times 10^5$) into the cerebellum of 6-week-old female NOD/SCID mice ($n = 6$ animals per group) (Charles River Laboratories) as previously described[66]. Brain tumour volume calculation was performed as described in ref. [66].

For allograft models, spontaneous MB from Ptch1$^{+/−}$ mice was isolated, minced, and pipetted to obtain a single-cell suspension, as previously described[67] and then infected with lentiviral particles. Equal volumes of cells ($2 \times 10^6$) were injected s.c. at the posterior flank of 6-week-old female BALB/c nude mice (nu/nu) ($n = 6$ animals per group) (Charles River Laboratories). Tumour growth was monitored by measuring the size by caliper and calculated as above describe. All animal experiments were approved by local ethic authorities and conducted in accordance with Italian Governing Law (D.lgs 26/2014; Prot. no. 03/2013). Animals were housed in the Institute's Animal Care Facilities, which meet international standards and were checked regularly by a certified veterinarian responsible for health monitoring, animal welfare supervision, and revision of experimental protocols and procedures.

**Immunohistochemistry**. The 4 µm thick sections were prepared from paraffin-embedded tissues and immunostained with anti-Gli1 or anti-Ki67 antibodies. After washes, secondary biotinylated antibodies were applied. Binding of antibodies was detected with the Mouse to Mouse HRP (DAB) Staining System (Scytek Laboratories, Inc., Logan, UT, USA) according to the manufacturer's protocol.

**Molecular dynamics**. The crystallographic structure of SuFuWT/Gli3 complex coded by PDB ID: 4BLD[68] was used as starting point in MD simulations. Particularly, chain A was selected, and structural gaps were filled. For larger portions such as the IDR disordered region, a preliminary folding experiment was performed starting from a linear sequence of residues. WT and mutant SuFu/Gli3 complexes were solvated in a rectilinear box of TIP3P water molecules buffering 12 Å from the protein. MD simulations were performed according to the procedure

described previously[66,69], but in this case unrestrained trajectories were initially equilibrated for 50 ns and then produced for 200 ns. Free energy of binding of Gli3 to SuFu WT and K321457R double mutant was estimated by the Molecular Mechanics Generalised Born Surface Area (MM-GBSA) approach[70].

**Statistical analysis**. Statistical analysis was performed using the StatView 4.1 software (Abacus Concepts, Berkeley, CA, USA). Statistical tests were appropriately chosen for each experiment. For in vivo studies statistical differences were analysed by Mann–Whitney $U$-test for non-parametric values and a $P < 0.05$ was considered significant. For all other experiments, $P$-values were determined using Student's $t$-test and statistical significance was set at $P < 0.05$. Results are expressed as mean ± SD from an appropriate number of experiments (at least three biological replicas).

**Data availability**. All data supporting the findings of this study are within the Article and Supplementary Files, or available from the authors upon request.

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

## Acknowledgements

We thank C. Brou for the gift of Itch$^{-/-}$ MEFs, R.Toftgard for the gift of SuFu$^{-/-}$ MEFs, R. Lefkowitz for the gift of β-arrestin2$^{-/-}$ MEFs, and G. Giannini for the critical discussion. This work was supported by Associazione Italiana Ricerca Cancro (AIRC) Grant IG14723, IG17575 and IG20801, PRIN 2012-2013 (2012C5YJSK002), Progetti di Ricerca di Università di Roma La Sapienza, Pasteur Institute/Cenci Bolognetti Foundation, Istituto Italiano di Tecnologia (IIT), Grant 111537 by the Deutsche Krebshilfe to M.K. and S.M.P. LLS was supported by PhD Degree Program in Biotechnology in clinical medicine, University of Rome La Sapienza.

## Author contributions

P.I., R.F., F.B., and L.D.M. conceived and designed the study. P.I., R.F., and F.B. performed most of the experiments. P.I., L.L.S., D.M., and M.S. generated the SuFu mutants and performed ubiquitylation in vitro or in vivo assays. P.I., F.B., R.A., M.P., and A.P. performed the animal experiments and analysis. A.G., D.G., S.C., G.C, M.M., E.D.S., E.F., C.C., M.M., I.S., M.K., and S.M.P. discussed the results, and provided critical reagents and comments. P.I., D.G., and L.D.M. wrote the manuscript.

## Additional information

**Competing interests:** The authors declare no competing interests.

