## [Peer Review File · Nature Communications]

Reviewers' comments:

Reviewer #1 (Remarks to the Author):

Infante and collaborators show that SUFU, a tumor suppressor in Sonic Hedgehog (SHH) signaling and medulloblastoma, interacts with the HECT E3 ubiquitin ligase Itch that mediates its ubiquitination on lysine 63 without affecting its half-life.

Using enforced expression of each of the proteins in the complex in HEK293T, the authors clearly demonstrate that Itch interacts with SUFU via two of its four WW domains and identify lysine 63 in SUFU to be required for Itch-dependent polyubiquitination. In lysates of whole mouse cerebella at different times after birth, ubiquitination of SUFU was maximal at P10, a time when most of the proliferation of granule neuron progenitors (GNPs) is complete suggesting that Itch-dependent SUFU polyubiquitination increases as cells exit cycle and differentiate to migrate into the IGL. β -arrestin2 but not β -arrestin1 co-precipitated with Sufu and Itch, although the amount of Itch in the complex was modest.

Because polyubiquitination of SUFU by Itch is required for the interaction of SUFU with GLI3 and the suppression of GLI3 transcriptional activity, they show that the SUFU K321/457R mutant no longer binds GLI3 with high affinity, leading to activation of transcription of SHH-dependent genes, including Gli1, Gli2 and cyclin D1 and cyclin D2 to drive proliferation. To demonstrate that the interaction of SUFU with GLI3 suppressed proliferation and tumor development, the authors used DAOY cells, an established medulloblastoma cell line with high MYC expression that is thought to be from a patient with a Group3 medulloblastoma. Enforced expression of the wild type SUFU inhibited the proliferation of DAOY cells in flank allografts whereas overexpression of the SUFU K321/457R mutant increased tumorigenesis.

Itch has been previously published to negatively regulate SUFU, however, these manuscripts have not identified Lysine 63 as the site of poly-ubiquitination that does not lead to proteasomal degradation. The suppression of GLI3 and other GLI proteins, Gli1 and Gli2, by SUFU is not entirely novel and has been demonstrated previously as correctly referred by the authors. Thus it is completely expected that the loss of interaction between GLI3 and SUFU would activate SHH-dependent gene transcription.

Major criticism

Most of the experiments are performed in established cell lines by enforced expression of SUFU wild type and specific mutants, Itch and β -arrestin2. Although biochemical experiments and ubiquitination experiments with mutants require ectopic expression in easily transfectable HEK293T and MEF cells, the interaction of Itch with β -arrestin2 and SUFU, as well as the increase of the Sufu/Gli3 complex should be validated in granule neuron progenitors, the cell of origin of SHH medulloblastoma.

The authors attempted to do this in mouse cerebella. Based on the data presented here, I would expect the Itch/ β -arrestin2/ Sufu complex to be detected in cerebella and GNPs early after birth and decrease as cells differentiate. This complex should be absent in SHH tumor cells. Many mouse models of SHH medulloblastoma are available and should be used to validate the results.

I found the use of human DAOY cells that do not recapitulate human SHH medulloblastoma questionable. Just because it is a human medulloblastoma cell line does not entirely justify its use. Eventhough enforced expression of the SUFU K321/457R mutant increased tumorigenesis and migration of DAOY cells, we know that the DAOY cells themselves give tumors in recipient mice. What is the tumorigenic potential of DAOY cells on their own without enforced expression of SUFU or the lysine mutants? Would any medulloblastoma line behave similarly? A mouse model of SHH medulloblastoma that recapitulate the human disease should be used instead.

Other criticisms

1. Some of the panels in Figure 2b seem to have been inverted. Endogenous Itch should not be present in Itch^{-/-} MEFs. Instead the levels are extremely high. The authors argue that ubiquitination of SUFU by Itch does not affect SUFU degradation, yes in Figure 2e the reblot with anti-FLAG shows a significant reduction of SUFU levels when Itch is overexpressed. This seemingly unexpected result should be explained in the text.
2. Figure 3 panel e shows a substantial amount of ubiquitination of the lysine less SUFU. This results is inconsistent with the data presented in Figure 2d.
3. To demonstrate the levels of SUFU ubiquitination during normal development, the authors used whole lysates of mouse cerebella at different times after birth. Although I agree that SUFU ubiquitination should increase with decreased number of proliferating GNP, the immunoblot shows that SuFu levels decrease as ubiquitination increases which is not what is expected if SUFU ubiquitination by Itch does not affect its degradation. At P12 when ubiquitination is low, SUFU levels are very low to increase again at P15 whereas the total SuFu levels are high. These data are confusing and should be explained. The authors should provide the levels of endogenous Itch and β arrestin2 protein levels during cerebellar development as well as the levels of Gli2 and Gli3, not just Gli1.
4. Figure 5d, the levels of Itch in total lysates should be shown for all lanes, not just the first two to demonstrate overexpression of Itch.
5. Data in Figure S3 should be introduced into the Results section when lysine mutants of SUFU are first used in Figure 2. I think that the schematic of SUFU wild type and mutants will be helpful to the readers.

Reviewer #2 (Remarks to the Author):

The authors carried out a biochemical screen to identify E-3 ligases that can ubiquitinate the Hedgehog pathway protein SuFu. In this screen they identified Itch, which had been previously implicated in the pathway as a regulator of the Hedgehog receptor Patched-1. The authors show that Itch interacts with SuFu through WW domains 1 and 2, and can poly-ubiquitinate SuFu on lysine residues K321 and K457 using a K63 linkage. They argue that ubiquitination at these sites increase SuFu binding to Gli3 promoting formation of the Gli3 repressor. They go on to show that beta-Arrestin2 promotes ubiquitination of SuFu by Itch and that a human medulloblastoma cell line overexpressing the K321/457R SuFu mutation was more aggressive than one overexpressing wild type SuFu. The paper is clearly written and addresses an important regulatory node in the Hedgehog pathway.

This is a potentially interesting paper, though several issues need to be addressed.

The pattern of SuFu ubiquitination in MEF cells is rather different than that when Itch is overexpressed. Is poly-ubiquitination the relevant event or are mono and di-ubiquitination the normal modifications?

A more detailed description of the immunoblotting would be appropriate especially where levels are quantified. Are signals in the linear range?

The effects of the K321/457R mutations are interpreted as being a consequence of failure to ubiquitinate. This need not be the case. The interaction with Gli3 could be disrupted just as a consequence of the amino acid changes. The same is true for the effects on tumorigenesis. The experiments in figure 3f are not compelling as overexpression of Itch modestly increases the interaction of wild type SuFu with Gli3 and there is a similar trend with K321/457R SuFu. The authors might want to perform this experiment with Itch siRNA knockdown, though the effect in figure 3d was not dramatic. It could also be done in a coupled transcription/translation system.

The phenotypic effects of ubiquitination are interpreted as a consequence of decreased interaction with Gli3. This would be predicted to depend upon the relative amounts of ubiquitinated and non-ubiquitinated SuFu. What fraction of SuFu is ubiquitinated and is this sufficient to account for the effects? Unlike ubiquitination as a transient step in the degradation of a protein, this modification would be required to be present in a significant fraction of the proteins to have an effect.

Reviewer #3 (Remarks to the Author):

In this study the authors explore the molecular basis for the tonic interaction between Gli3 and SuFu, which leads to the proteasome-dependent processing of Gli3 to the repressor Gli3R and to stabilization of Gli3FL. Strong evidence is provided that the ubiquitination of SuFu by the WW domain HECT E3, Itch, in the absence of activation of Hh signaling, is critical for Gli3R processing. These findings establish a role for non-proteasomal ubiquitination of SuFu, presumably through generation of K63-linked chains on two specific Lys (K321 and K457), as a key regulatory mechanism for transcriptional repression via the Hh pathway in the absence of active signaling. In addition, the study includes data supporting a role for β -arrestin2 (β arr2), which had been implicated as a modulator of Hh signaling, as being integral to Itch-SuFu interactions. Itch-mediated ubiquitination is indirectly implicated as a means of limiting cell growth both in vitro and in xenografts through evaluation of SuFu K321/457R. Of particular interest is that mutations that include loss of these Lys are found in medulloblastoma.

The central findings are provocative and potentially of high impact and important for understanding the regulation of a developmentally and clinically important signaling pathway. In particular, the study establishes an additional, non-degradative, role for the ubiquitin system in Hh signaling. The experiments are generally of high quality and the writing is clear. However, as presented, the study is incomplete. With appropriate revisions and addressing of points raised below, this reviewer believes it would warrant publication.

1. The critical endpoint being regulated by the ubiquitination of SuFu via Itch is enhanced association of Gli3 with SuFu that, according to the authors and the cited reference, should allow for the proteasomal processing of Gli3 to the repressive Gli3R and the stabilization of Gli3FL, preventing it from undergoing complete proteasomal degradation. Therefore a prediction of expressing a form of SuFu incapable of being ubiquitinated by Itch should be both a decrease in the rate of formation of Gli3R and an increase in the rate of degradation of Gli3FL. The former seems to be the case in the snapshot in Fig. 3i, however, the interpretation of this experiment is itself confusing as the bookkeeping between the cytoplasmic Gli3FL and the nuclear Gli3R doesn't add up and is not explained. In this regard, it is important when showing Gli3FL and Gli3R that they be shown on the same blot and exposure and that the percent of nuclear and cytoplasmic fractions used be made clear and be equivalent, if at all possible. The only evidence suggesting the loss of Gli3FL with K371/457R SuFu is the single point IP in 3G. However, it is also evident based on Fig. 4A that the authors' IPs don't necessarily agree with their assessment of whole cell lysates - a point of some concern. As both the generation of Gli3R and stability of Gli3FL are critical points essential to the physiological relevance of this study, the authors need to provide convincing data that WT SuFu, but not Lys mutant SuFu, leads to the generation of Gli3R and the stabilization of Gli3FL - preventing its proteasomal degradation. These studies must be bona fide assessments of stability by cycloheximide chase or preferably by pulse-chase metabolic labeling (ensuring that their IPs are quantitative or at least of equal efficiency). Further, these experiments are best accomplished by knockdown of endogenous SuFu and re-expression of either the WT or K371/457R forms of SuFu. Related to this, issues of effects on levels of Gli3 need to be considered in interpretation of 3f. It would seem that if the levels of Gli3FL are increased by Itch, that overall Gli3FL-SuFu association would increase by mass action.

2. Related to the previous point, once the importance of K371 and K457 are established in the

study, the importance of Itch is largely extrapolated based on the double mutation, with the assumption that the only function of these Lys are to serve as Itch substrates. Given that the Itch-SuFu-Gli3 axis at the heart of the paper, critical studies related to Gli3 and to biological outcomes should be corroborated by manipulation of Itch. Therefore, the stability assessment noted above should also be carried out with Itch knockdown followed by re-expression of either WT or inactive and non-interacting Itch (WW1WW2 double mutant). Similarly, select cell growth and tumor studies should include experiments that utilize both WT and mutant Itch and WT and mutant SuFu in a combinatorial manner compared to appropriate controls, preferably by knockdown and re-expression rather than overexpression. A central prediction is that effects of the K371/457R double mutant should be relatively immune from manipulation of Itch, whereas the WT should not be.

3. The data on the role of β -arrestin 2 (β arr2) and its role in Itch-SuFu interactions in Fig. 5 would not appear to be interpretable without additional information. In particular, there is no direct evidence that this reviewer can discern from the manuscript, that the three proteins exist in a complex and that this complex facilitates Itch-SuFu interactions as opposed to Itch activity in general. Further, the use of SAG as a surrogate for loss of β arr2 from the complex would need to be clearly established for interpretation of these results. A superior approach would be to knockdown β arr2 (and re-express as a control) to directly demonstrate a role in complex formation between Itch and SuFu, although this would not rule out the possibility that β arr2 alters Itch and indirectly leads to a stabilization of the Itch-SuFu interaction. Other experiments (immunodepletion, re-IP etc.) would be necessary to establish that the three truly exist as a complex. It would also be important in establishing the authors' point to show that β arr2 plays a role in *in vitro* ubiquitination. In this regard, it is curious that the SuFu ubiquitination is so impressive in the absence of added β arr2. This could be due to the type of lysate being used in these *in vitro* translations and whether these contain endogenous β arr2. This itself requires clarification (see below).

4. Depending on the type of *in vitro* translation system used, the authors need to exercise care in implying a direct association between Itch and SuFu. Eukaryotic translation system (and particularly rabbit retics) can surely have proteins that can serve as adaptors. This is especially important to consider as the SuFu binding site for Itch is not identified or analyzed in this study.

5. The family of WW domain HECT E3s contains a lot of similarities and there are examples of redundancy among these E3s. Given this, the authors should assess which WW domains are most similar to WW1 and WW2 of Itch and, utilizing ligase(s) containing these, establish whether there is the potential for redundancy in this system both through *in vitro* and cellular assays. Even if the Itch^{-/-} MEFs and knockdowns have an effect, this does not mean that other ligases, in different cell types, may not have an important role depending on their relative expression. As with the assessment of stability, such experiments need to be carried out using multiple amounts of such ligases and in cellular experiments this assessment would be most informative in cells in which Itch is either knocked down or out.

6. While the *in vitro* evidence for K63 chains would favor a non-degradative function for Itch, Itch and other E3s that form K63 chains *in vitro* are known to target proteins for degradation, and the data on SuFu levels with increasing Itch in 3B is negative data. Thus, the authors should provide at least one time course experiment, with appropriate controls, demonstrating that Itch fails to affect the stability of endogenous SuFu.

7. The authors extrapolate from the *in vitro* ubiquitination assay that SuFu is modified on K63. Certainly the data presented would lead to such an interpretation (particularly if there is no degradation - previous point). However, this *in vitro* assessment is incomplete as it only addresses K48 and K63 linkages. To be complete, this reviewer suggests two alternative types of studies. The first would be to assess a complete panel of Lys mutants of ubiquitin and show that K63R uniquely inhibits polyubiquitination. The second, and preferable, approach would be to overexpress either K63R or K48R ubiquitin in cells and demonstrate that the former, but not the latter, results

in a decrease in chain length.

8. While the evidence indirectly points to ubiquitination of SuFu substantially increasing association with Gli3, no direct evidence that the associated SuFu is ubiquitinated relative to the pool of free SuFu. Experiments should be carried out to attempt to demonstrate, by Gli3 vs. SuFu IP, that there is evidence for such ubiquitination.

Other points

9. The figure legends and methods collectively are incomplete. Reference has already been made to the translation system; there is no information on many of the plasmids and recombinant proteins used in this study.

10. In Figure 1A what does the bottom panel represent (presumably different E3s) and how should we interpret this? Are they really almost identical in molecular weight? How were they detected? This is a bit sloppy.

11. Figure 2b bottom panel presumably represents total not "endog." Itch.

12. Interpretation of the K321R mutation by itself is not interpretable given the low level of SuFu in 2e. As this is the only assay in which the single mutant is assessed, and therefore implicated as a target, a better quality experiment is required.

13. As noted above, the relative amounts of SuFu in Fig. 4a in the IPs vs. lysates suggests an experiment that is technically suboptimal. If this is due to some characteristic of the IP antibody it should be explained to the reader.

14. In 4c is there a technical reason why alterations in levels of Gli3FL and Gli3R are not shown with SAG treatment?

Re: Revised Manuscript NCOMMS-16-11710-T.

"Itch/ β arrestin2-dependent non-proteolytic ubiquitylation of SuFu controls Hedgehog signalling and medulloblastoma tumourigenesis" by Infante et al.

Point-by-point response to Reviewers' comments:

Reviewer #1 (Remarks to the Author):

Reviewer: Infante and collaborators show that SUFU, a tumor suppressor in Sonic Hedgehog (SHH) signaling and medulloblastoma, interacts with the HECT E3 ubiquitin ligase Itch that mediates its ubiquitination on lysine 63 without affecting its half-life.

Using enforced expression of each of the proteins in the complex in HEK293T, the authors clearly demonstrate that Itch interacts with SUFU via two of its four WW domains and identify lysine 63 in SUFU to be required for Itch-dependent polyubiquitination. In lysates of whole mouse cerebella at different times after birth, ubiquitination of SUFU was maximal at P10, a time when most of the proliferation of granule neuron progenitors (GNPs) is complete suggesting that Itch-dependent SUFU polyubiquitination increases as cells exit cycle and differentiate to migrate into the IGL. β -arrestin2 but not β -arrestin1 co-precipitated with Sufu and Itch, although the amount of Itch in the complex was modest.

Because polyubiquitination of SUFU by Itch is required for the interaction of SUFU with GLI3 and the suppression of GLI3 transcriptional activity, they show that the SUFU K321/457R mutant no longer binds GLI3 with high affinity, leading to activation of transcription of SHH-dependent genes, including Gli1, Gli2 and cyclin D1 and cyclin D2 to drive proliferation. To demonstrate that the interaction of SUFU with GLI3 suppressed proliferation and tumor development, the authors used DAOY cells, an established medulloblastoma cell line with high MYC expression that is thought to be from a patient with a Group3 medulloblastoma. Enforced expression of the wild type SUFU inhibited the proliferation of DAOY cells in flank allografts whereas overexpression of the SUFU K321/457R mutant increased tumorigenesis.

Itch has been previously published to negatively regulate SUFU, however, these manuscripts have not identified Lysine 63 as the site of poly-ubiquitination that does not lead to proteosomal degradation. The suppression of GLI3 and other GLI proteins, Gli1 and Gli2, by SUFU is not entirely novel and has been demonstrated previously as correctly referred by the authors. Thus it is completely expected that the loss of interaction between GLI3 and SUFU would activate SHH-dependent gene transcription.

Authors: We thank the Reviewer for the helpful comments on our work. The major concern of the Reviewer appears to be the lack of novelty, since he/she claims that "Itch has been previously published to negatively regulate SUFU" and that "The suppression of GLI3 and other GLI proteins by SUFU is not entirely novel". While others have partially addressed the functional and molecular interactions between SuFu and Gli proteins, the molecular mechanisms that regulate SuFu activity and in particular SuFu/Gli molecular interactions were still elusive, so far. In the present study we provide completely novel evidences on this issue:

1) we identified the mechanism controlling SuFu/Gli3 dynamic interaction via the Itch/ β -arrestin2-dependent and K63-linked polyubiquitination of SuFu;

2) while there is evidence that SuFu antagonizes Numb/Itch mediated degradation of Gli1 (*Lin C et al., Developmental Biology 2014*), we provide for the first time evidence that Itch directly controls SuFu;

3) Furthermore, we unveiled for the first time the role of β -arrestin2 in maintaining the Hh signalling off in the absence of ligand, by forming a complex with Itch and SuFu.

Thus, we believe that our data provide a significant progress in understanding the regulation of Hedgehog signaling, unveiling a new mechanism controlling the tumour suppressive function of SuFu and how its alteration contributes to medulloblastoma.

Major criticism:

Reviewer: Most of the experiments are performed in established cell lines by enforced expression of SUFU wild type and specific mutants, Itch and β -arrestin2. Although biochemical experiments and ubiquitination experiments with mutants require ectopic expression in easily transfectable HEK293T and MEF cells, the interaction of Itch with β -arrestin2 and SUFU, as well as the increase of the SuFu/Gli3 complex should be validated in granule neuron progenitors, the cell of origin of SHH medulloblastoma.

The authors attempted to do this in mouse cerebella. Based on the data presented here, I would expect the Itch/ β -arrestin2/Sufu complex to be detected in cerebella and GNPs early after birth and decrease as cells differentiate. This complex should be absent in SHH tumor cells. Many mouse models of SHH medulloblastoma are available and should be used to validate the results.

Authors: As suggested, we now demonstrate the interaction between SuFu/Gli3 and β -arrestin2/Itch/SuFu in a more physiological context. To this end, both the SuFu/Gli3 and the β -arrestin2/Itch/SuFu complexes were analyzed in postnatal cerebella tissues during development (P3-P7-P10-P15). In agreement with the levels of SuFu ubiquitination in mouse cerebella (**revised Figure 4a**), the formation of the SuFu/Gli3 (**new revised Figure 4c**) and β -arrestin2/Itch/SuFu complexes (**new revised Figure 5h**) increases in the second week after birth, as the pathway is progressively switched off.

As requested by the Reviewer we analyzed the Itch/SuFu/ β arrestin2 complex in primary tumors derived from *Ptch*^{+/-} mice (the most used mouse model of medulloblastoma). In this mouse model, medulloblastoma formation results from the deletion of the *Ptch* gene that leads to constitutive activation of the Hh pathway (Goodrich *et al.*, Science 1997).

As expected, compared to cerebellum tissue derived from 10 days postnatal mice (P10), in which the pathway is switched off, we observed a reduction of the Itch/ β arrestin2/SuFu complex formation in this tumor (**new revised Figure 5k**).

Reviewer: I found the use of human DAOY cells that do not recapitulate human SHH medulloblastoma questionable. Just because it is a human medulloblastoma cell line does not entirely justify its use. Eventhough enforced expression of the SUFU K321/457R mutant increased tumorigenesis and migration of DAOY cells, we know that the DAOY cells themselves give tumors in recipient mice. What is the tumorigenic potential of DAOY cells on their own without enforced expression of SUFU or the lysine mutants? Would any medulloblastoma line behave similarly? A mouse model of SHH medulloblastoma that recapitulate the human disease should be used instead.

Authors: The most appropriate way to address this issue would be having human SuFu mutation in a mouse model, which unfortunately does not exist. Therefore, to accomplish Reviewer's request we have now further validated the relevance of Itch-dependent SuFu ubiquitylation in primary medulloblastoma cells derived from *Ptch*^{+/-} mouse, where SHH pathway is constitutively activated. As expected, overexpression of WT SuFu in these cells inhibited tumor growth, consistently with the downstream action of SuFu on the Hh pathway. This effect was not observed upon the expression of the K321/457 SuFu mutant (**new revised Figure S7 and new revised Figure 8a-e**), consistent with the reduction of Itch/SuFu/ β arrestin2 complex in *Ptch*^{+/-} derived medulloblastoma, reported above (**new revised Figure 5k**).

The experiments performed on human medulloblastoma Daoy cells helped us to establish a proof-of-concept for our model, according to which, the impairment of the Itch-dependent SuFu ubiquitylation inhibits the oncosuppressive properties of SuFu.

We have used human medulloblastoma Daoy cells for the different reasons: i) Daoy cells belong to a Shh-MBs subgroup according to several authors (Triscott *et al.*, Cancer Res 73:6734, 2013; Ivanov *et al.*, JBiotechnology 236:10, 2016) including a classification based on the Nanostring nCounter system, which distinguishes medulloblastoma subgroups on the basis of the expression

of the 22 established medulloblastoma subtyping genes (Northcott et al., *Acta Neuropathol* 123:615, 2012). Therefore, Daoy cells are not classified as a Group 3-MB subtype, have not Myc or MycN amplification and do not express high levels of Myc (Triscott et al., *Cancer Res* 73:6734, 2013; Ivanov et al., *JBiotechnology* 236:10, 2016), as we have further confirmed by immunoblotting (please see figure below); ii) their growth is indeed dependent on Hh signalling (Ferretti *EMBO J* 27:2616, 2008) and iii) their treatment with Hh antagonists leads to growth inhibition and a decrease in Gli1 and other Hh targets mRNA levels (Canettieri et al., *Nat Cell Biol* 2010; Bar et al., *Am J Pathol.* 170:347, 2007).

D283: Group 3-medulloblastoma cell line; DAOY: Shh-medulloblastoma cell line; HCT116: colorectal cancer cell line; HEK293T: human embryonic kidneys cell line; CHLA: Group 4-medulloblastoma cell line.

We agree with the Referee that the Daoy cells themselves generate tumors in recipient mice. Indeed, the experiments shown in old Figures 6 and 7 (**new revised Figures 6 and 7**) were originally performed also including control group of mice either engrafted or implanted with Daoy infected with control lentiviruses (**new revised Figures 6 and 7, respectively**). We have now included results from all experimental groups (CTR, SuFu WT, SuFu mutant). As expected, compared with control, the expression of wild type SuFu decreased the tumour growth rate of Daoy cells and the tumor mass volume. Strikingly, the K321/457R mutations, which render SuFu insensitive to Itch-dependent ubiquitylation, completely abrogated the oncosuppressive functions of SuFu (**new revised Figure 6 and Figure 7**).

Other criticisms

Reviewer: 1. Some of the panels in Figure 2b seem to have been inverted. Endogenous Itch should not be present in Itch^{-/-} MEFs. Instead the levels are extremely high.

Authors: We apologize if Figure 2b was misleading due to an unclear labeling. We have now re-labelled the figure panels. Exogenous wild type Itch and the Itch mutant were detected with an anti-Flag antibody (lanes 3 and 4); endogenous Itch was detected with an anti-Itch antibody (lanes 1 and 2), which detects also the exogenous protein, thereby justifying the high levels of Itch in lanes 3 and 4 (**new revised Figure 2b**).

Reviewer: The authors argue that ubiquitination of SUFU by Itch does not affect SUFU degradation, yet in Figure 2e the reblot with anti-FLAG shows a significant reduction of SUFU levels when Itch is overexpressed. This seemingly unexpected result should be explained in the text.

Authors: The panel shown in that figure represents the levels of immunoprecipitated proteins and likely reflects a difference in the efficiency of immunoprecipitation. In the revised version, we have added a panel showing that the levels of SuFu in the whole cell extracts are not affected by Itch expression (**new revised Figure S2**). Nevertheless, we have replaced the previous experiment with another one of a better quality (**new revised Figure 2f**).

Reviewer: 2. Figure 3 panel e shows a substantial amount of ubiquitination of the lysine less SUFU. This results is inconsistent with the data presented in Figure 2d.

Authors: The panel shown in former figure 3e (**now Figure 2m**) indicates that Itch catalyzes the assembly of K63-linked polyubiquitin chains. To demonstrate this, we used several ubiquitin mutants, including lysine-less ubiquitin, not lysine-less SuFu, which was instead used in the experiment shown in Figure 2d. The pattern shown in the lane of K-less ubiquitin is due to presence of endogenous ubiquitin (present in wheat germ extract) that competes with recombinant ubiquitin used for the in vitro reaction. To further demonstrate that Itch polyubiquitylates SuFu through lysine 63-mediated linkages, we used a ubiquitin mutant in which K63 is replaced by arginine (K63R). In this case, the K63R mutant was not linked to SuFu. Indeed, we observed a significant decrease in the formation of high polyubiquitin chains.

Reviewer: 3. To demonstrate the levels of SUFU ubiquitination during normal development, the authors used whole lysates of mouse cerebella at different times after birth. Although I agree that SUFU ubiquitination should increase with decreased number of proliferating GNPs, the immunoblot shows that Sufu levels decrease as ubiquitination increases which is not what is expected if SUFU ubiquitination by Itch does not affect its degradation. At P12 when ubiquitination is low, SUFU levels are very low to increase again at P15 whereas the total Sufu levels are high. These data are confusing and should be explained.

Authors: The panel shown in that figure represents the levels of immunoprecipitated SuFu (Reblot: SuFu) and is likely to reflect a difference in the efficiency of immunoprecipitation. The bottom panels show that the levels of SuFu in the whole cell extract at the P12 time point are not affected by ubiquitylation (**revised Figure 4a**).

Reviewer: The authors should provide the levels of endogenous Itch and β arrestin2 protein levels during cerebellar development as well as the levels of Gli2 and Gli3, not just Gli1.

Authors: We now show the levels of the proteins as requested by the Reviewer (**revised Figure 4a**).

Reviewer: 4. Figure 5d, the levels of Itch in total lysates should be shown for all lanes, not just the first two to demonstrate overexpression of Itch.

Authors: These data have been added in the revised manuscript (**new revised Figure 5e**)

Reviewer: 5. Data in Figure S3 should be introduced into the Results section when lysine mutants of SUFU are first used in Figure 2. I think that the schematic of SUFU wild type and mutants will be helpful to the readers.

Authors: A schematic of SUFU is now shown in Figure 2 (**new revised Figure 2e**).

Reviewer #2 (Remarks to the Author):

Reviewer: The authors carried out a biochemical screen to identify E-3 ligases that can ubiquitinate the Hedgehog pathway protein SuFu. In this screen they identified Itch, which had been previously implicated in the pathway as a regulator of the Hedgehog receptor Patched-1. The

authors show that Itch interacts with SuFu through WW domains 1 and 2, and can poly-ubiquitinate SuFu on lysine residues K321 and K457 using a K63 linkage. They argue that ubiquitination at these sites increase SuFu binding to Gli3 promoting formation of the Gli3 repressor. They go on to show that beta-Arrestin2 promotes ubiquitination of SuFu by Itch and that a human medulloblastoma cell line overexpressing the K321/457R SuFu mutation was more aggressive than one overexpressing wild type SuFu. The paper is clearly written and addresses an important regulatory node in the Hedgehog pathway.

This is a potentially interesting paper, though several issues need to be addressed.

Authors: We thank the Reviewer for the positive general comments on our study.

Reviewer: The pattern of SuFu ubiquitination in MEF cells is rather different than that when Itch is overexpressed. Is poly-ubiquitination the relevant event or are mono and di-ubiquitination the normal modifications?

Authors: Poly-ubiquitylation is the predominant and relevant ubiquitylation of SuFu as shown in the postnatal cerebellum tissues during development in revised Figure 4a, which demonstrates the correlation between SuFu ubiquitylation and the activity of Hh signalling. Mono and di-ubiquitination modifications are usually intermediates forms and can be detected under physiological conditions. It is expected that ectopic expression of an E3 ligase, such as Itch, strongly increases the efficiency of ubiquitylation of its own substrates resulting in the formation of long ubiquitin chains and, hence, in the accumulation of the poly-ubiquitylated SuFu.

Reviewer: A more detailed description of the immunoblotting would be appropriate especially where levels are quantified. Are signals in the linear range?

Authors: We apologize for lack of clarity. In the revised manuscript, we now provide a detailed description of the immunoblotting. Since films have a limited linear range for light detection, it is crucial to determine the appropriate film exposure time. To assess whether the image exposure is within a linear dynamic range, the film has been exposed to the blot at different times (30sec, 1min, 2min, 3min, 5min). The most intense protein band on the film has been identified. The intensity of the band has then been read on all images and plotted against the exposure time. The selected image was the one in which the most intense band was in the linear range.

Reviewer: The effects of the K321/457R mutations are interpreted as being a consequence of failure to ubiquitinate. This need not be the case. The interaction with Gli3 could be disrupted just as a consequence of the amino acid changes. The same is true for the effects on tumorigenesis.

Authors: Despite a plethora of published reports used the replacement of lysine with arginine to analyze the requirement of lysine residues for ubiquitin conjugation, we agree with the reviewer that the K321/457R mutations might affect the Gli3-SuFu interaction just as a consequence of the amino acid change. To monitor the possible structural consequence of the K321/457R mutations as well as its impact on Gli3 binding affinity, we performed computational studies based on molecular dynamics (MD) simulations and free energy of binding calculations. In our computational studies, we used the available crystallographic structure of SuFu in complex with a Gli3 peptide (PDB ID: 4BLD) that well represents a static snapshot of the SuFu/Gli3 interaction at high resolution (2.8 Å). The structure of full-length SuFu was rebuilt by filling structural gaps. Afterwards, conformational features of Gli3 binding site on SuFu wild-type (SuFuWT) and the double mutant K321/457R (SuFuK321/457R) were analyzed along 200 ns of unrestrained MD simulations. A representative structure of SuFuWT/Gli3 and SuFuK321/457R-Gli3 systems was then extracted from MD trajectories and used for structural analysis. Notably, as showed in **new Supplementary Figure S4 a and b**, the overall structure and organization of the Gli3 binding site

in SuFu is not affected by the double K/R mutation. This is not particularly surprising if we consider that K321 and K457 are solvent exposed as well as far away from the Gli3 binding site in X-ray crystal structures and computational MD structures as well. Moreover, lysine and arginine share the same electronic and shape features, thus suggesting that K/R mutation should not impact on the structure of a protein. Instead, their different chemical properties may account for different reactivity and functions. From a thermodynamic standpoint, the free energy of Gli3 binding to SuFu (WT or K321/457R mutant) was calculated by the Molecular Mechanics Generalized Born Surface Area (MM-GBSA) approach, which is a well-established method to estimate the relative affinity of two binding partners along MD trajectories. Results showed in the **Table of new revised Figure S4** clearly highlight that the interaction between Gli3 and SuFu is not affected by the K321/K457 double mutation.

In conclusion, this computational study indicates that the interaction between Gli3 and SuFu is not expected to be impaired by the K321/457R mutation of SuFu either at the structural or thermodynamics level.

Reviewer: The experiments in figure 3f are not compelling as overexpression of Itch modestly increases the interaction of wild type SuFu with Gli3 and there is a similar trend with K321/457R SuFu. The authors might want to perform this experiment with Itch siRNA knockdown, though the effect in figure 3d was not dramatic. It could also be done in a coupled transcription/translation system.

Authors: As suggested, we have now performed a new experiment in which we assessed the role of Itch-dependent SuFu ubiquitylation in the formation of SuFu/Gli3 complex in the absence of Itch. Since the Reviewer finds modest the effect of Itch siRNA, despite significant and reproducible (middle panel, figure 3d), we have performed this experiment in Itch^{-/-} MEFs. As shown in **new revised Figure 3a**, overexpression of Itch significantly increases the interaction of Gli3 with wild type SuFu, but not with SuFuK321/457R.

Reviewer: The phenotypic effects of ubiquitination are interpreted as a consequence of decreased interaction with Gli3. This would be predicted to depend upon the relative amounts of ubiquitinated and non-ubiquitinated SuFu. What fraction of SuFu is ubiquitinated and is this sufficient to account for the effects? Unlike ubiquitination as a transient step in the degradation of a protein, this modification would be required to be present in a significant fraction of the proteins to have an effect.

Authors: The reviewer is correct since the quantification of the ubiquitylated fraction of SuFu may provide further proof of our model. However, we have to consider that:

- ubiquitin chain extension by K63 linkages, as well as by other K-linkages, is a dynamic process since polyubiquitinated forms are unstable and transient and the substrate's polyubiquitylated signal can be remodeled by several deubiquitinating enzymes (DUBs) within the cells;

- the quantification of the ubiquitylated fraction of SuFu would need to be performed on the endogenous protein and one should be able to inhibit the DUBs present in the cell lysates. The latter are known to work non specifically in vitro and very difficult to inhibit. This makes the detection of ubiquitylated species present in cells so challenging and, in fact, to our knowledge a successful and precise quantification of the relative amounts of ubiquitylated and non-ubiquitylated species has not been performed for any substrate.

Moreover, whatever amount of SuFu would be found ubiquitylated, we would not be able to experimentally address whether this amount would be sufficient or insufficient for the biological effect we observe.

Nevertheless, our findings strongly indicate that Itch-dependent SuFu ubiquitylation, regulating Gli3 protein, is required for the suppressive function of SuFu in Hh signaling and its oncosuppressive properties. Indeed, we observed that silencing Itch decreases the ubiquitylation of SuFu. We have demonstrated that the Itch-dependent ubiquitylation of SuFu affects the interaction of SuFu with Gli3. In fact, we have identified the ubiquitylated lysine residues (K321/457) of SuFu and

demonstrated by both NanoBiT and co-IP assays that a K321/457R mutant, no longer ubiquitinated by Itch, displays a decreased ability to bind Gli3 when compared to wild type SuFu (**new revised Figure 3a, d and e**). Further, we have shown that the expression of the SuFuK321/457R mutant leads to a reduction of Gli3 protein levels and Gli3FL stability when compared to wild type SuFu (**new revised Figure 3f,g,h**). Moreover, we have shown that overexpression of Gli3 and wild type SuFu in SuFu^{-/-} MEFs caused a significant reduction of mRNA levels of Hh target genes and that this effect was rescued in presence of SuFuK321/457R mutant (**new revised Fig. 3j**). Finally, we have demonstrated that SuFuK321/457R mutation counteracts SuFu oncosuppressive function both in vitro and in vivo (**new Figure 6, 7, 8 and Supplementary Figures S5, S6, S7**).

Reviewer #3 (Remarks to the Author):

In this study the authors explore the molecular basis for the tonic interaction between Gli3 and SuFu, which leads to the proteasome-dependent processing of Gli3 to the repressor Gli3R and to stabilization of Gli3FL. Strong evidence is provided that the ubiquitination of SuFu by the WW domain HECT E3, Itch, in the absence of activation of Hh signaling, is critical for Gli3R processing. These findings establish a role for non-proteasomal ubiquitination of SuFu, presumably through generation of K63-linked chains on two specific Lys (K321 and K457), as a key regulatory mechanism for transcriptional repression via the Hh pathway in the absence of active signaling. In addition, the study includes data supporting a role for β -arrestin2 (β arr2), which had been implicated as a modulator of Hh signaling, as being integral to Itch-SuFu interactions. Itch-mediated ubiquitination is indirectly implicated as a means of limiting cell growth both in vitro and in xenografts through evaluation of SuFu K321/457R. Of particular interest is that mutations that include loss of these Lys are found in medulloblastoma.

The central findings are provocative and potentially of high impact and important for understanding the regulation of a developmentally and clinically important signaling pathway. In particular, the study establishes an additional, non-degradative, role for the ubiquitin system in Hh signaling. The experiments are generally of high quality and the writing is clear. However, as presented, the study is incomplete. With appropriate revisions and addressing of points raised below, this reviewer believes it would warrant publication.

Authors: We thank the Reviewer for appreciating the importance and soundness of our data.

1) Reviewer. The critical endpoint being regulated by the ubiquitination of SuFu via Itch is enhanced association of Gli3 with SuFu that, according to the authors and the cited reference, should allow for the proteasomal processing of Gli3 to the repressive Gli3R and the stabilization of Gli3FL, preventing it from undergoing complete proteasomal degradation. Therefore a prediction of expressing a form of SuFu incapable of being ubiquitinated by Itch should be both a decrease in the rate of formation of Gli3R and an increase in the rate of degradation of Gli3FL. The former seems to be the case in the snapshot in Fig. 3i, however, the interpretation of this experiment is itself confusing as the bookkeeping between the cytoplasmic Gli3FL and the nuclear Gli3R doesn't add up and is not explained. In this regard, it is important when showing Gli3FL and Gli3R that they be shown on the same blot and exposure and that the percent of nuclear and cytoplasmic fractions used be made clear and be equivalent, if at all possible.

The only evidence suggesting the loss of Gli3FL with K371/457R SuFu is the single point IP in 3G. However, it is also evident based on Fig. 4A that the authors IPs don't necessarily agree with their assessment of whole cell lysates - a point of some concern. As both the generation of Gli3R and stability of Gli3FL are critical points essential to the physiological relevance of this study, the authors need to provide convincing data that WT SuFu, but not Lys mutant SuFu, leads to the generation Gli3R and the stabilization of Gli3FL - preventing its proteasomal degradation. These studies must be bona fide assessments of stability by cycloheximide chase or preferably by pulse-chase metabolic labeling (ensuring that their IPs are quantitative or at least of equal efficiency).

Further, these experiments are best accomplished by knockdown of endogenous SuFu and re-expression of either the WT or K371/457R forms of SuFu.

Related to this, issues of effects on levels of Gli3 need to be considered in interpretation of 3f. It would seem that if the levels of Gli3FL are increased by Itch, that overall Gli3FL-SuFu association would increase by mass action.

Authors: We apologize for lack of clarity, due to space limit, and for having presented confusing data. We prefer to carry out distinct immunoblots of Gli3FL and Gli3R because the Gli3 antibody we used (R&D #AF3690) often recognizes non specific bands that make difficult the interpretation of data. We have now performed new subcellular fractionation assays, according to a previous report (Humke et al., 2010), where intracellular compartmentalization of Gli3FL and Gli3R in the presence of SuFu WT or SuFuK321/457R mutant was analyzed on the same blot (**New revised Figure 3h and revised Methods**). An equal percentage of nuclear and cytoplasmic fractions were taken to ensure that equal amounts of each fraction were loaded on the gel, although, we cannot rule out unaccounted losses during fractionation due to multiple washing and centrifugation steps. As shown in the new revised Figure 3h, the expression of the SuFu K321/457R mutant results in a reduction of Gli3FL and Gli3R levels.

As suggested by the Reviewer, to address the role of Itch-dependent SuFu ubiquitylation in the regulation of Gli3FL stability and Gli3R generation, first we have verified if the steady-state levels of Gli3FL and Gli3R are affected by expression of SuFu wild type or SuFu lysine mutant in SuFu^{-/-} MEF cells. To this end, SuFu^{-/-} cells were infected with control lentivirus or lentiviruses expressing wild type SuFu or SuFu mutant. As previously reported by Humke et al., 2010, compared with control cells, the steady-state levels of Gli3FL and Gli3R were much higher in SuFu^{-/-} cells overexpressing SuFu wild type. However, we observed that the presence of SuFu lysine mutant impaired the accumulation of both forms of Gli3 (**new revised Figure 3f**).

Next, as suggested by Reviewer, we have performed cycloheximide chase experiments in SuFu^{-/-} MEFs following re-expression of wild type SuFu or the SuFu lysine mutant. As shown in **new revised Figure 3g**, the half-life of Gli3FL was shorter in the presence of the SuFu mutant than in the presence of wild type SuFu. Since SuFu controls the rate of Gli3R production and not the rate of its degradation (Humke et al., 2010), the half-life of Gli3R was unchanged in the presence of SuFu mutant (**new revised Figure 3g**).

2) Reviewer. Related to the previous point, once the importance of K371 and K457 are established in the study, the importance of Itch is largely extrapolated based on the double mutation, with the assumption that the only function of these Lys are to serve as Itch substrates. Given that the Itch-SuFu-Gli3 axis at the heart of the paper, critical studies related to Gli3 and to biological outcomes should be corroborated by manipulation of Itch. Therefore, the stability assessment noted above should also be carried out with Itch knockdown followed by re-expression of either WT of inactive and non-interacting Itch (WW1WW2 double mutant).

Authors: As suggested, we have performed cycloheximide chase experiments in Itch^{-/-} MEFs following re-expression of wild type or the inactive Itch mutant. We observed that the half-life of Gli3 was shorter in the absence of Itch or in the presence of the ItchC830A catalytically inactive mutant when compared to the one in the presence of wild type Itch (**new revised Figure 3i**). Our results indicate that Itch, by inducing SuFu ubiquitylation, increases SuFu/Gli3 interaction, thus leading to Gli3 stabilization.

Reviewer. Similarly, select cell growth and tumor studies should include experiments that utilize both WT and mutant Itch and WT and mutant SuFu in a combinatorial manner compared to appropriate controls, preferably by knockdown and re-expression rather than overexpression. A central prediction is that effects of the K371/457R double mutant should be relatively immune from manipulation of Itch, whereas the WT should not be.

Authors: We have performed the requested experiments that are now shown in **new revised Figure S5, panels e and f**. The BrdU proliferation assay indeed showed that the SuFuK321/457R

mutant was unable to affect the growth of Daoy cells following Itch or SuFu modulation.

3) Reviewer. The data on the role of β -arrestin 2 (β arr2) and its role in Itch-SuFu interactions in Fig. 5 would not appear to be interpretable without additional information. In particular, there is no direct evidence that this reviewer can discern from the manuscript, that the three proteins exist in a complex and that this complex facilitates Itch-SuFu interactions as opposed to Itch activity in general. Further, the use of SAG as a surrogate for loss of β arr2 from the complex would need to be clearly established for interpretation of these results. A superior approach would be to knockdown β arr2 (and re-express as a control) to directly demonstrate a role in complex formation between Itch and SuFu, although this would not rule out the possibility that β arr2 alters Itch and indirectly leads to a stabilization of the Itch-SuFu interaction. Other experiments (immunodepletion, re-IP etc.) would be necessary to establish that the three truly exist as a complex.

Authors: To address these questions, we performed the following experiments: i) we tested if the three proteins, β arr2, SuFu and Itch exist as a ternary complex. To this purpose, MEFs were transfected with plasmids expressing Flag-Itch, GFP- β arr2 and HA-SuFu. Protein lysates were immunoprecipitated with anti-Flag agarose beads. One-third of anti-Flag IP volume (IP) was recovered for polyacrilammide gel run, while two-thirds have been eluted with Flag-peptide and then re-immunoprecipitated (re-IP) with HA-agarose beads. Immunoblots shown in **new revised Figure 5f** revealed that the three proteins bind one another both in the IP and re-IP, demonstrating that they are assembled in a ternary complex. (ii) We tested the role of β -arrestin2 in the formation of the Itch/SuFu complex by co-immunoprecipitation in β arr2^{-/-} MEFs. The interaction of SuFu with Itch significantly increases following re-expression of β -arrestin2 (**new revised Figure 5g**).

Reviewer. It would also be important in establishing the authors' point to show that β arr2 plays a role in *in vitro* ubiquitination. In this regard, it is curious that the SuFu ubiquitination is so impressive in the absence of added β arr2. This could be due to the type of lysate being used in these *in vitro* translations and whether these contain endogenous β arr2. This itself requires clarification (see below).

Authors: We performed the requested experiment, which is now shown in **revised Figure 5c**. The presence of purified recombinant β arr2 increased Itch-induced SuFu ubiquitylation in a reconstituted *in vitro* ubiquitylation system containing ubiquitin, E1, E2 (UbcH7), ATP and *in vitro* synthesized radiolabelled [³⁵S] SuFu as substrate. This result indicates that β arr2 stimulates the ubiquitylation of SuFu *in vitro* in the presence of recombinant Itch.

As suggested by the Reviewer, the strong ubiquitylation of SuFu observed in the absence of added β arr2 could be due to the presence of endogenous β arr2 in wheat germ extract used for the *in vitro* translation (TnT[®] Coupled Wheat Germ Extract System, Promega, Madison, WI, USA). Moreover, this is likely due to the intrinsic catalytic activity of Itch, which can either function on its own or in conjunction with accessory or adaptor proteins.

4) Reviewer. Depending on the type of *in vitro* translation system used, the authors need to exercise care in implying a direct association between Itch and SuFu. Eukaryotic translations system (and particularly rabbit retics) can surely have proteins that can serve as adaptors. This is especially important to consider as the SuFu binding site for Itch is not identified or analyzed in this study.

Authors: As requested, we have demonstrated the direct binding of Itch with SuFu in a system in which only recombinant proteins (SuFu and Itch) were used as is now shown in **revised Figure 1f**. A GST-pull down assay was performed in the presence of purified recombinant Itch and GST-SuFu or GST alone used as negative control.

5) Reviewer. The family of WW domain HECT E3s contains a lot of similarities and there are examples of redundancy among these E3s. Given this, the authors should assess which WW domains are most similar to WW1 and WW2 of Itch and, utilizing ligase(s) containing these, establish whether there is the potential for redundancy in this system both through *in vitro* and cellular assays. Even if the Itch^{-/-} MEFs and knockdowns have an effect, this does not mean that other ligases, in different cell types, may not have an important role depending on their relative expression. As with the assessment of stability, such experiments need to be carried out using multiple amounts of such ligases and in cellular experiments this assessment would be most informative in cells in which Itch is either knocked down or out.

Authors: Itch belongs to the neuronal precursor cell-expressed developmentally downregulated 4 (NEDD4)/Rsp5p-like HECT family of E3 ubiquitin ligases, which share a characteristic modular structure, with an N-terminal lipid interacting C2 region, four WW protein–protein interacting motifs and a C-terminal catalytic HECT domain (Liu, 2004). There are nine Nedd4 members in mammals: Nedd4, Nedd4L, Itch, WWP1, WWP2, Smurf1, Smurf2, NEDL1 and NEDL2 (Rotin and Kumar, Nature 2009). Phylogenetically, Itch clusters with WWP1 and WWP2, while Nedd4-1 and 2, Smurfs, and NEDL1 and 2 form separate clusters. Since the high homology (50-85%) of WWP1, Nedd4 and Smurf with the WW1 e WW2 domains of Itch, we investigated the effect of these E3 ligases on SuFu ubiquitylation. As reported in **new revised Figure S1**, the selected E3 ligases failed to promote SuFu ubiquitylation, as also previously shown in Figure 1a, at least for Nedd4.

We have performed these experiments in easily transfectable HEK293T cells as we also did for the majority of the *in vivo* ubiquitylation assays presented in this study. However, we cannot rule out that, in other cellular contexts and/or under specific stimuli, SuFu could be ubiquitylated by other members of the WW domain HECT E3 ligases.

6) Reviewer. While the *in vitro* evidence for K63 chains would favor a non-degradative function for Itch, Itch and other E3s that form K63 chains *in vitro* are known to target proteins for degradation, and the data on SuFu levels with increasing Itch in 3B is negative data. Thus, the authors should provide at least one time course experiment, with appropriate controls, demonstrating that Itch fails to affect the stability of endogenous SuFu.

Authors: To investigate whether Itch influences SuFu stability, we performed cycloheximide pulse experiments in wild type MEF vs Itch^{-/-} MEFs. The stability of the endogenous SuFu is similar either in the absence or presence of Itch (**new revised Figure 2I**). These results indicate that Itch does not affect SuFu stability.

7) Reviewer. The authors extrapolate from the *in vitro* ubiquitination assay that SuFu is modified on K63. Certainly the data presented would lead to such an interpretation (particularly if there is no degradation - previous point). However, this *in vitro* assessment is incomplete as it only addresses K48 and K63 linkages. To be complete, this reviewer suggests two alternative types of studies. The first would be to assess a complete panel of Lys mutants of ubiquitin and show that K63R uniquely inhibits polyubiquitination. The second, and preferable, approach would be to overexpress either K63R or K48R ubiquitin in cells and demonstrate that the former, but not the latter, results in a decrease in chain length.

Authors: We have performed the requested experiment. As shown in **new revised Figure S3**, the expression of K63R ubiquitin, but not K48R ubiquitin, strongly reduces the Itch-dependent ubiquitylation of SuFu. In light of the data also presented in response to point 6) (no degradation of SuFu), we believe we have now provided strong evidences for our claim.

8) Reviewer. While the evidence indirectly points to ubiquitination of SuFu substantially increasing association with Gli3, no direct evidence that the associated SuFu is ubiquitinated relative to the

pool of free SuFu. Experiments should be carried out to attempt to demonstrate, by Gli3 vs. SuFu IP, that there is evidence for such ubiquitination.

Authors: To address this point, we have performed two types of experiments. First, we have carried out a co-IP followed by re-IP experiments, where cell lysates from MEFs transfected with Flag-Gli3, HA-SuFu, myc-Ub and Itch were immunoprecipitated with an anti-Flag Gli3 antibody. Then, the immunocomplexes, containing Gli3 and its interactors, were subjected to re-immunoprecipitation with an anti-HA SuFu antibody, followed by immunoblotting with an anti-HA or anti-Myc antibody to detect conjugated SuFu-Ub, and with an anti-Flag antibody to verify the binding of HA-purified SuFu with Gli3. This experiment indicates that SuFu co-purifying with Gli3 is ubiquitylated (**new revised Figure 3d**).

Second, we have performed a ubiquitylation assay in vivo, where HA-tagged SuFu was expressed in MEFs alone or in presence of Gli3 or Itch or both. Cell lysates were immunoprecipitated with an anti-HA antibody, followed by immunoblotting with an anti-Myc antibody to detect conjugated Myc-ubiquitin. We observed a strong increase in SuFu ubiquitylation when both Gli3 and Itch were present, and this was associated with an increased interaction of SuFu with Gli3 (**new revised Figure 3e**).

Other points

Reviewer 9. The figure legends and methods collectively are incomplete. Reference has already been made to the translation system; there is no information on many of the plasmids and recombinant proteins used in this study.

Authors: We have included all the requested information and clarifications.

Reviewer 10. In Figure 1A what does the bottom panel represent (presumably different E3s) and how should we interpret this? Are they really almost identical in molecular weight? How were they detected? This is a bit sloppy.

Authors: The bottom panel represents the expression levels of the different E3 ligases. This information has been clarified in the **revised Figure 1a**.

Reviewer 11. Figure 2b bottom panel presumably represents total not "endog." Itch.

Authors: The blot is now labeled properly (**revised Figure 2b**).

Reviewer 12. Interpretation of the K321R mutation by itself is not interpretable given the low level of SuFu in 2e. As this is the only assay in which the single mutant is assessed, and therefore implicated as a target, a better quality experiment is required.

Authors: We have replaced the previous experiment with another one where the levels of immunoprecipitated SuFu (Reblot: Flag-SuFu) are identical in all lanes (**revised Figure 2f**). The result is the same shown previously, as we observe a significant reduction in the ubiquitylation of SuFuK321R as well as the SuFuK457R mutant (when compared to wild type SuFu).

Reviewer 13. As noted above, the relative amounts of SuFu in Fig. 4a in the IPs vs. lysates suggests an experiment that is technically suboptimal. If this is due to some characteristic of the IP antibody it should be explained to the reader.

Authors: The panel shown in that figure represents the levels of immunoprecipitated SuFu

(Reblot: SuFu) and we agree with the reviewer is likely to reflect a difference in the efficiency of immunoprecipitation. In the figure is also present a panel showing that the levels of SuFu in the whole cell extract are unchanged during cerebellar development (**revised Figure 4a**).

Reviewer 14. In 4c is there a technical reason why alterations in levels of Gli3FL and Gli3R are not shown with SAG treatment?

Authors: The levels of Gli3FL and Gli3R are now shown in **new revised Figure 4d**.

Reviewers' comments:

Reviewer #2 (Remarks to the Author):

The resubmitted manuscript is substantially improved from the previous submission. The data in the figures are of high quality and well controlled. The work identifies an interesting additional level of regulation for SuFu, a key component of the Hedgehog signal transduction cascade and implicates this regulation in MB tumor progression. My previous concerns have largely been addressed. My only suggestion would be to add a bit more detail to the panel descriptions in the figures.

Reviewer #3 (Remarks to the Author):

This revision is greatly improved compared to the original submission. The main weakness remains that data supporting the proposed mechanism of how ubiquitination of SuFu by Itch regulates Gli3.

As pointed out in the original review, it is not clear that K371/457R fails to interact with Gli3 because of loss of ubiquitination. This can be assessed by binding experiments using purified recombinant SuFu proteins. Based on the proposed model, the generation of Gli3R will be reduced by SuFu K327/457R mutation. In addition, the stability of Gli3FL will be reduced in the presence of SuFu K327/457R. The authors tried to address this possibility in Fig 3. Although Fig 3h shows reduced accumulation of Gli3FL and Gli3R in cells expressing SuFu mutant, the stability of Gli3FL is not affected as evidenced in Fig 3g (despite claim to the contrary in the manuscript text). Interestingly, Gli3R is present in SuFu KO cells suggesting SuFu is not needed for Gli3R generation. Furthermore, expression of either WT and mutant SuFu stabilizes Gli3FL and increases Gli3R in these cells. Unfortunately, the low quality of data in Fig 3i makes it difficult to assess the effect of Itch expression on the stability of Gli3FL.

The role of b-arrestin 2 (barr2) in SuFu-Itch interaction is not clear. In Fig 5f, the authors tried to establish the presence of a ternary complex. In this experiment, IP for Itch was followed by re-IP for SuFu. It is not clear what percentage of input is shown. However, the ratio of the different proteins in the IP compared to that in the re-IP will be consistent with a model in which Itch can interact with SuFu in the absence of barr2. The lack of controls in Fig 5e and 5g makes it difficult to interpret the data shown in these panels. Given the small effect of barr2 on Itch-dependent ubiquitination of SuFu in vitro, it will be useful to assess ubiquitination of SuFu in Itch KO MEFs. Similarly, the requirement for Itch-dependent ubiquitination for SuFu-Gli3 interaction should be assessed in Itch KO cells.

Reviewer #4 (Remarks to the Author):

The authors have added new data that in a satisfactory manner clarifies issues raised and substantiate findings and conclusions.

Re: Revised Manuscript NCOMMS-16-11710-T.

"Itch/ β arrestin2-dependent non-proteolytic ubiquitylation of SuFu controls Hedgehog signalling and medulloblastoma tumourigenesis" by Infante et al.

Point-by-point response to Reviewers' comments:

Reviewer #2

"The resubmitted manuscript is substantially improved from the previous submission. The data in the figures are of high quality and well controlled. The work identifies an interesting additional level of regulation for SuFu, a key component of the Hedgehog signal transduction cascade and implicates this regulation in MB tumor progression. My previous concerns have largely been addressed. My only suggestion would be to add a bit more detail to the panel descriptions in the figures".

Authors: We thank the Reviewer for appreciating the importance and soundness of our work. We now added more details to the panel descriptions in the figures, as suggested by Reviewer.

Reviewer #4

"The authors have added new data that in a satisfactory manner clarifies issues raised and substantiate findings and conclusions."

Authors: We thank the Reviewer for the positive comments on our revised manuscript.

Reviewer #3 (Remarks to the Author):

"This revision is greatly improved compared to the original submission. The main weakness remains that data supporting the proposed mechanism of how ubiquitination of SuFu by Itch regulates Gli3".

Authors: We are glad that the Reviewer believes that our paper is greatly improved. We have added in the re-revised paper new data addressing the Reviewer's criticisms and improved the presentation of the results as specified below.

As pointed out in the original review, it is not clear that K371/457R fails to interact with Gli3 because of loss of ubiquitination. This can be assessed by binding experiments using purified recombinant SuFu proteins. Based on the proposed model, the generation of Gli3R will be reduced by SuFu K327/457R mutation. In addition, the stability of Gli3FL will be reduced in the presence of SuFu K327/457R. The authors tried to address this possibility in Fig 3. Although Fig 3h shows reduced accumulation of Gli3FL and Gli3R in cells expressing SuFu mutant, the stability of Gli3FL is not affected as evidenced in Fig 3g (despite claim to the contrary in the manuscript text). Interestingly, Gli3R is present in SuFu KO cells suggesting SuFu is not needed for Gli3R generation. Furthermore, expression of either WT and mutant SuFu stabilizes Gli3FL and increases Gli3R in these cells. Unfortunately, the low quality of data in Fig 3i makes it difficult to assess the effect of Itch expression on the stability of Gli3FL."

Authors: In the first revision (Supplementary Figure S4) we have demonstrated that the effect of K321/457R mutations on the interaction between Gli3 and SuFu is a not a consequence of the nature of the substituted amino acids, but rather of loss of ubiquitination. Unfortunately, the use of recombinant proteins to assess whether the interaction is affected or not by ubiquitylation processes doesn't seem to be suitable. Indeed, recombinant proteins produced with standard approaches do not undergo poly-ubiquitylation and thus cannot be used to address this point. The generation of synthetic poly-ubiquitinated proteins and in vitro analysis of processing events would require instead a series of multiple and complicated approaches (e.g. generation and purification of in vitro-ubiquitylated SuFu, purification of full length Gli3, binding assays with in-vitro modified proteins, reconstitution of the proteasomal system to assess Gli3 processing, etc). Most of these procedures to our knowledge are unprecedented and their combination would most likely provide artifactual results.

To demonstrate that Itch-dependent ubiquitylation of SuFu regulates Gli3, we have shown that Gli3FL, and consequently the amount of Gli3R (**new revised Figure 3j and Figure 3l**), are reduced by the expression of the SuFu K321/457R mutant.

The endogenous levels of Gli3 (both Gli3 full-length and Gli3 repressor) are known to be greatly reduced and barely detectable in the absence of SuFu. In particular, Gli3FL is known to be much more labile than Gli3R in the absence of SuFu. Indeed, steady-state levels of both forms are very low, and sometimes undetectable in SuFu^{-/-} total cell lysates (see Fig. 3j) and even upon ectopic expression of SuFu (Chen M et al., G&D 2009; Jia et al., DevBiol 2009; Humke et al., G&D 2010; and our experiment in Figure 3j). For this reason, to examine the effect of SuFu K321/457R expression on the stability of Gli3FL, we measured the half-life of Gli3 in SuFu^{-/-} MEFs reconstituted with wild type SuFu or the SuFu K321/457R mutant. Only with this approach (previously utilized by Humke et al., G&D 2010) we could reliably detect and measure the stability of Gli3FL and Gli3R in SuFu^{-/-} MEFs cells (**new revised Figure 3k**).

As reported by Humke et al., 2010, re-expression of SuFu in SuFu^{-/-} cells results in the accumulation of both forms of Gli3. However, while the steady-state levels of Gli3FL and Gli3R and the stability of Gli3FL were higher in SuFu^{-/-} MEFs re-expressing ectopic SuFu (**new revised Figure 3j and 3k**), the half-life of Gli3R remained unchanged, in agreement with previous studies demonstrating that SuFu potentiates the formation of Gli3R by controlling the rate of Gli3R production but not of its degradation.

We also observed that, although the expression of either WT or the SuFu mutant stabilizes Gli3FL and increases Gli3R in SuFu^{-/-} MEFs, the half-life of Gli3FL was shorter in the presence of the K321/457R mutant compared to wild type SuFu. However, we cannot rule out the involvement of other mechanisms that may contribute to the accumulation of both forms of Gli3 by regulating SuFu activity independently of Itch and SuFu K321/457R mutations.

We apologize for the quality of the Figure 3i in the previous version of the manuscript. Itch^{-/-} MEFs are difficult to transfect or infect, and Gli3 antibody often recognizes non-specific bands, thus resulting in images with high background and of modest quality. We have replaced the previous experiment with one of improved quality (**new revised Figure 3n**). As shown in the previous version of the manuscript, the half-life of Gli3FL was shorter in the absence of Itch or in the presence of the ItchC830A catalytically inactive mutant compared to wild type Itch.

The role of b-arrestin 2 (barr2) in SuFu-Itch interaction is not clear. In Fig 5f, the authors tried to establish the presence of a ternary complex. In this experiment, IP for Itch was followed by re-IP for SuFu. It is not clear what percentage of input is shown. However, the ratio of the different proteins in the IP compared to that in the re-IP will be consistent with a model in which Itch can interact with SuFu in the absence of barr2. The lack of controls in Fig 5e and 5g makes it difficult to interpret the data shown in these panels.

Authors: To improve the data concerning the presence of a ternary complex and to clarify the role of β -arrestin 2 in the SuFu/Itch complex, we have performed a new experiment (**new revised Figure 5f**), in which tagged SuFu, Itch and β -arrestin 2 were expressed in WT MEFs in different combinations. In the first IP, Flag-Itch co-immunopurified with both GFP- β -arr2 and HA-SuFu. In the second IP anti-Flag immunoprecipitates were eluted with Flag peptide and re-precipitated with an anti-HA antibody. Again, all three proteins were detected by immunoblotting (**new revised Figure 5f**) indicating that SuFu, Itch and β -arr2 form a trimeric complex (5% of input was loaded). To help interpreting the data, we have also added the requested controls in Figure 5e and 5g (**new revised Figure 5e and 5g**). Taken together these results indicate that expression of β -arrestin 2 results in an increased interaction between SuFu and Itch (**new revised Figures 5f and 5g**).

Given the small effect of barr2 on Itch-dependent ubiquitination of SuFu in vitro, it will be useful to assess ubiquitination of SuFu in Itch KO MEFs.

Authors: We have now performed a ubiquitylation assay in Itch^{-/-} MEFs after ectopic expression of Itch alone or Itch together with β -arrestin 2 (**new revised Figure S5**).

Altogether our results indicate that the increased interaction between SuFu and Itch enhances the ubiquitylation of SuFu mediated by Itch and stabilizes the poly-ubiquitin chains, as illustrated by the accumulation of long polyubiquitin forms shown in *in vitro* ubiquitylation assay (Figure 5c; the quantification of poly-ubiquitylated SuFu of Figure 5c is shown below).

Similarly, the requirement for Itch-dependent ubiquitination for SuFu-Gli3 interaction should be assessed in Itch KO cells.

Authors: The requirement for Itch-dependent ubiquitination for SuFu-Gli3 interaction has now been investigated in Itch KO cells, as requested (**new revised Figures 3f,g**). SuFu/Gli3 binding was significantly increased after re-expression of Itch in Itch^{-/-} MEFs compared to control cells. Accordingly, SuFu/Gli3 binding was strongly reduced in WT MEFs after depletion of Itch by RNAi compared to control cells (**new revised Figures 3d,e**). These data support the role of Itch-dependent ubiquitination for SuFu/Gli3 interaction.

REVIEWERS' COMMENTS:

Reviewer #5 (Remarks to the Author):

The revised manuscript is substantially improved. The authors have provided new data that address this reviewer's concerns.